# Travel surveillance uncovers dengue virus dynamics and introductions in the Caribbean

Emma Taylor-Salmon[1,2,20] ✉, Verity Hill[2,20], Lauren M. Paul [3,20], Robert T. Koch[2,20], Mallery I. Breban[2], Chrispin Chaguza[2], Afeez Sodeinde[2], Joshua L. Warren[4,5], Sylvia Bunch[6], Natalia Cano[6], Marshall Cone[6], Sarah Eysoldt[6], Alezaundra Garcia[6], Nicadia Gilles[6], Andrew Hagy[6], Lea Heberlein[6], Rayah Jaber[6], Elizabeth Kassens[6], Pamela Colarusso[7], Amanda Davis[7], Samantha Baudin[8], Edhelene Rico[8], Álvaro Mejía-Echeverri[8], Blake Scott[9], Danielle Stanek[9], Rebecca Zimler [9], Jorge L. Muñoz-Jordán[10], Gilberto A. Santiago[10], Laura E. Adams[10], Gabriela Paz-Bailey[10], Melanie Spillane[11,12], Volha Katebi[11], Robert Paulino-Ramírez[13], Sayira Mueses[13], Armando Peguero[13], Nelissa Sánchez[13], Francesca F. Norman [14], Juan-Carlos Galán [15], Ralph Huits [16], Davidson H. Hamer[17], Chantal B. F. Vogels [2,18,21], Andrea Morrison [9,21] ✉, Scott F. Michael [3,21] ✉ & Nathan D. Grubaugh [2,5,18,19,21] ✉

Dengue is the most prevalent mosquito-borne viral disease in humans, and cases are continuing to rise globally. In particular, islands in the Caribbean have experienced more frequent outbreaks, and all four dengue virus (DENV) serotypes have been reported in the region, leading to hyperendemicity and increased rates of severe disease. However, there is significant variability regarding virus surveillance and reporting between islands, making it difficult to obtain an accurate understanding of the epidemiological patterns in the Caribbean. To investigate this, we used travel surveillance and genomic epidemiology to reconstruct outbreak dynamics, DENV serotype turnover, and patterns of spread within the region from 2009-2022. We uncovered two recent DENV-3 introductions from Asia, one of which resulted in a large outbreak in Cuba, which was previously under-reported. We also show that while outbreaks can be synchronized between islands, they are often caused by different serotypes. Our study highlights the importance of surveillance of infected travelers to provide a snapshot of local introductions and transmission in areas with limited local surveillance and suggests that the recent DENV-3 introductions may pose a major public health threat in the region.

Dengue is an acute febrile illness viral infection caused by dengue virus (DENV), which is transmitted to humans through the bite of an infected *Aedes* mosquito[1–3]. There are four genetically- and antigenically-related DENV serotypes, and while infection with one serotype generally produces long-lasting homotypic immunity, it only induces short-lived heterotypic immunity, after which infection with another serotype may result in more severe disease[1,2,4]. Almost half of the world's population lives in dengue-endemic areas, with children bearing the majority of the

A full list of affiliations appears at the end of the paper. ✉e-mail: emma.taylor-salmon@yale.edu; andrea.Morrison@flhealth.gov; smichael@fgcu.edu; nathan.grubaugh@yale.edu

disease burden[5,6]. Dengue has evolved from a sporadic disease into a major public health threat in the Americas, with reported case numbers reaching 20 million in the last 5 years in 46 countries[7]. The Caribbean has often been a source of reintroduction and spread of all four DENV serotypes over the last thirty years[8,9]. Two other *Aedes*-borne viruses also emerged in this region: chikungunya virus in 2013[10,11] and Zika virus in 2016[12,13]. Local infectious disease surveillance and reporting is essential for public health efforts regarding outbreak response and disease mitigation; however, the control of outbreaks is challenging in low- and middle-income countries where resources are limited[14]. In addition, dengue outbreaks can overwhelm health systems, further weakening surveillance capacity - particularly in countries that may not have strong laboratory infrastructure[15-17]. Therefore, the areas where disease surveillance is the most needed - like the Caribbean - are often the same areas where we have limited publicly available data.

Infectious disease surveillance of travelers has been shown to supplement local surveillance in low-resource areas[18-24]. Infected travelers can also be sentinels for pathogen transmission in locations where outbreaks have not yet been reported[25]. For example, characterization of pathogens in infected travelers was used to reconstruct the global spread of H1N1 in 2009[26] and an unreported Zika outbreak in Cuba in 2017[18]. Utilizing traveler data could be particularly useful for supplementing surveillance in the Caribbean because of the popularity of tourism in the region[27]. In the United States, dengue is the leading cause of febrile illness among travelers returning from the Caribbean[23,25,28-30]. In Florida, the number of travel-associated dengue cases has increased dramatically in recent years[31]. Florida is a good sentinel for the Caribbean, due to its geographic location and high volume of travel back and forth between Florida and the islands. Therefore, we hypothesize that we can use surveillance of dengue-infected travelers diagnosed in Florida who recently returned from the Caribbean to better reconstruct DENV dynamics in the region.

In this study, we combined surveillance and sequencing of DENV from infected travelers to estimate local outbreak sizes, track serotypes, and map the patterns of lineage spread from 2009 to 2022. We determined that travel-associated infections closely mirrored local rates for countries and territories with robust local surveillance. An increase in infected travelers returning from Cuba in 2022 has previously been reported[32], leading to the assumption of a large dengue outbreak that was not reported to Pan-American Health Organization (PAHO), the primary source for information regarding dengue spread in the Americas[7]. We used surveillance data among travelers to estimate that the 2022 outbreak in Cuba was similar in size to other large outbreaks reported in the Americas. We used serotype data to decipher inter-island differences during outbreak years. By sequencing DENV isolates from travel associated cases, we uncovered two recent DENV-3 introductions from Asia into Jamaica and Cuba, the latter of which has now been detected in several other locations in the region[33]. Overall, our study highlights the importance of dengue surveillance among travelers and genomic epidemiology in supplementing local infectious disease surveillance in resource-limited locations by elucidating DENV transmission and spread within the Caribbean.

## Results

### Dengue travel cases to supplement local surveillance

Dengue incidence in the Americas rose significantly over the last four decades, reemerging as a major public health concern and culminating in ~17.5 million reported cases from 2010 to 2019 and ~20 million cases since 2019[7]. This coincided with a rapid increase in travel-associated dengue cases within the United States, especially in Florida. The Florida Department of Health (FDOH) has a robust dengue surveillance system, which captures symptomatic cases reported in patients who traveled to a dengue-endemic area within the two weeks prior to illness onset (Table S1). From 2009 to 2022, the FDOH reported between 19 (in 2017) to 929 (in 2022) travel-associated dengue cases per year, for a total of 2300 cases, with the majority (1815 cases, 78.9%) occurring in travelers who recently returned from the Caribbean (Supplementary Fig. S1). Overall, four countries and one US territory made up 75.6% (1737 cases) of all travel-associated dengue cases reported in Florida from 2009 to 2022: Cuba, Puerto Rico, Dominican Republic, Haiti, and Jamaica (Fig. 1). There is significant variability in local dengue surveillance and reporting among Caribbean countries and territories, which can impact local outbreak responses and travel advisories[34,35]. We hypothesize that we can address this variability by using dengue surveillance of infected travelers in Florida to detect surveillance gaps within these five Caribbean islands.

To determine the extent to which travel-associated cases reported in Florida correlated with local reporting in the Caribbean, we compared local and travel-associated dengue case numbers from 2009 to 2022 (Fig. 2). We obtained yearly suspected and confirmed dengue cases reported by Cuba, Dominican Republic, Haiti, Jamaica, and Puerto Rico from PAHO and all reports of travel-associated dengue cases from the FDOH. First, we compared the local and travel case trends for each country or territory. We found significant positive correlations from the Dominican Republic (Pearson $r = 0.764$, $p = 0.001$), Jamaica ($r = 0.960$, $p < 0.001$), and Puerto Rico ($r = 0.911$, $p < 0.001$), but the data from Cuba ($r = 0.496$, $p = 0.070$) and Haiti ($r = -0.694$, $p = 0.056$) were not correlated (Fig. 2A). We then used flight data from the United States Department of Transportation (Supplementary Fig. S2) to estimate the 'travel infection rates' as the number of travel-associated cases per 100,000 air passenger journeys into Florida per year from each location[18,36]. Comparing the travel infection rates to the local rates (dengue cases per 100,000 population), we again found positive correlations from the Dominican Republic (Pearson $r = 0.750$, $p = 0.002$), Jamaica ($r = 0.912$, $p < 0.001$), and Puerto Rico ($r = 0.935$, $p < 0.001$), but not from Cuba ($r = 0.458$, $p = 0.1$) and Haiti ($r = -0.825$, $p = 0.012$; Fig. 2B). The strong correlations between travel and local cases from the Dominican Republic, Jamaica, and Puerto Rico are similar to what we previously estimated for Zika[18] and others estimated for dengue[36], reflecting the robust and consistent arbovirus surveillance systems at these locations. Notably, we found peaks of travel-associated infections from Cuba in 2022 that were not captured by the dengue case data reported by PAHO (Fig. 2A). This increase in travel-associated cases was also seen in other parts of the world[32]. Our data also show disagreements between travel and local dengue cases in Haiti, specifically spikes in travel-associated cases in 2012 and 2018 and a spike in local cases in 2021 that were not reflected in the other data (Fig. 2A). In fact, there are many years of missing dengue case data in the PAHO repository from Haiti, and these gaps are frequently paired with natural disasters (e.g. 2010 earthquake). Overall, we demonstrate here that dengue travel surveillance in Florida can help supplement local surveillance in the Caribbean region to infer relative local case dynamics.

### Estimating local dengue infection rates using travelers

Having demonstrated that the Dominican Republic, Jamaica, and Puerto Rico have significant positive correlations between local and travel-associated dengue infection rates (travel cases per 100,000 air passenger journeys; Fig. 2), we then created a negative binomial regression model using these relationships to estimate local infection rates from Cuba and Haiti. We estimate that the 2022 dengue outbreak in Cuba is larger than any detected on the island since at least 2010 and similar in size to other large outbreaks found throughout the Americas (Fig. 3).

In areas with inadequate local case reporting, previous studies suggested that sentinel surveillance of infected travelers can be used to estimate virus transmission dynamics and spread from endemic locations[18,26,36]. Only 3036 laboratory-confirmed dengue cases were reported by Cuba in 2022, similar to what was reported in 2019[7]. This is inconsistent with the number of cases we detected from Cuba using travel surveillance (Figs. 1 and 2A). Moreover, dengue case data from

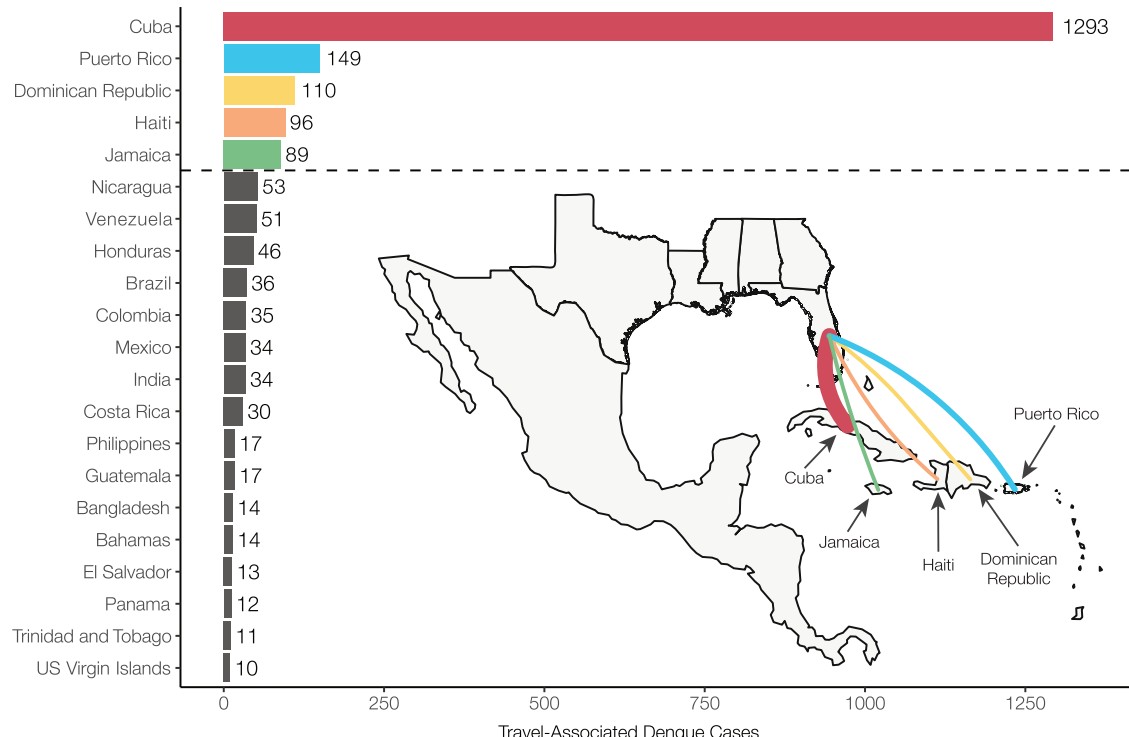

**Fig. 1 | Cuba, Dominican Republic, Haiti, Jamaica, and Puerto Rico make up the majority of travel-associated dengue cases reported in Florida from 2009 to 2022.** Countries and territories are listed by total number of travel cases for each inferred origin of infection based on travel history, in descending order. Only countries or territories with at least 10 associated travel infections are shown. The complete data can be found in Supplementary Table S1. The inset shows a map of the location of the top 5 associated country origins of travel cases reported in Florida, with the line width proportional to the number of travel cases.

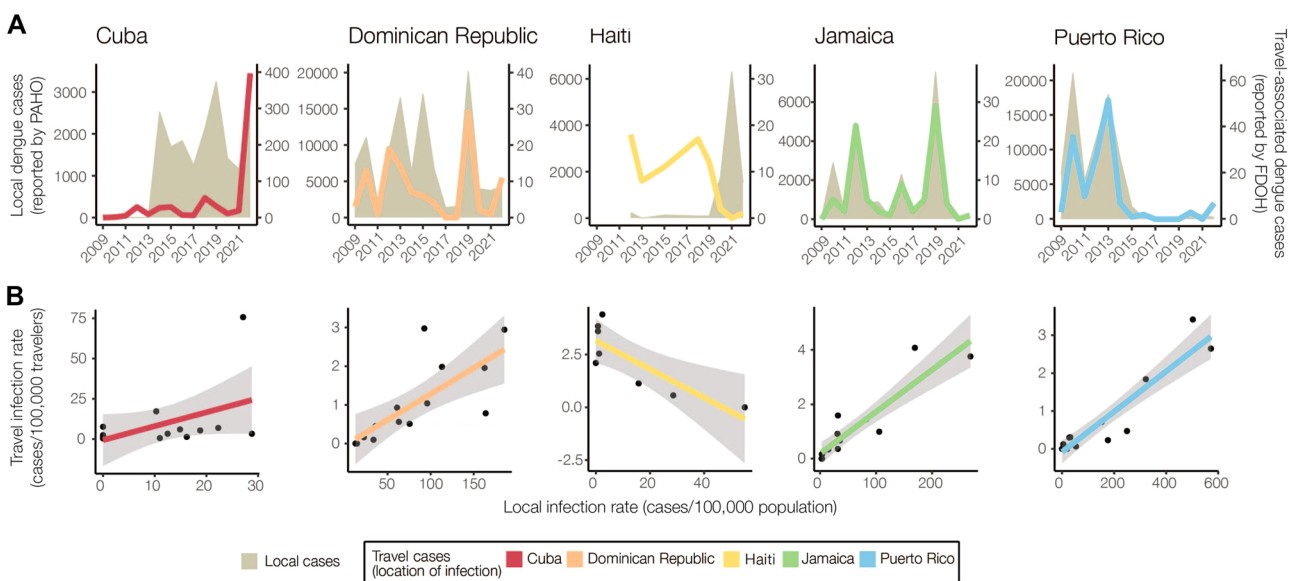

**Fig. 2 | Travel surveillance from Florida correlates with local dengue infection rates in endemic countries and territories with robust local surveillance and case reporting. A** Yearly local dengue cases (left y-axis, gray shaded area) reported by PAHO and yearly travel-associated dengue cases (right y-axis, colored lines) reported by FDOH were sorted by the origin of exposure. The datasets were compared using Pearson's correlation coefficient. There were strong positive correlations between travel and local cases for the Dominican Republic (Pearson $r = 0.764$, $p = 0.001$), Jamaica ($r = 0.960$, $p < 0.001$), and Puerto Rico ($r = 0.911$, $p < 0.001$), with no significant correlation for Cuba ($r = 0.498$, $p = 0.070$) and a negative correlation for Haiti ($r = -0.694$, $p = 0.056$). **B** The local dengue virus incidence rates for each country or territory were calculated by the number of locally reported cases per month per 100,000 population. The travel dengue virus incidence rates for each country or territory of presumed exposure were calculated by the number of travel-associated cases per month per 100,000 air passenger journeys entering Florida from endemic locations. Colored lines represent a two-sided linear regression model with local infection rate as the predictor variable and travel infection rate as the outcome variable. Gray shaded area represents standard error. There were strong positive correlations between travel and local incidence for the Dominican Republic (Pearson $r = 0.750$, $p = 0.002$), Jamaica ($r = 0.912$, $p < 0.001$), and Puerto Rico ($r = 0.935$, $p < 0.001$), with no significant correlation for Cuba ($r = 0.458$, $p = 0.100$) and a negative correlation for Haiti ($r = -0.825$, $p = 0.012$). The negative correlation between the local and travel infection rates may have been driven by a decreased travel volume to Florida from 2020 to 2022 (Supplementary Fig. S2).

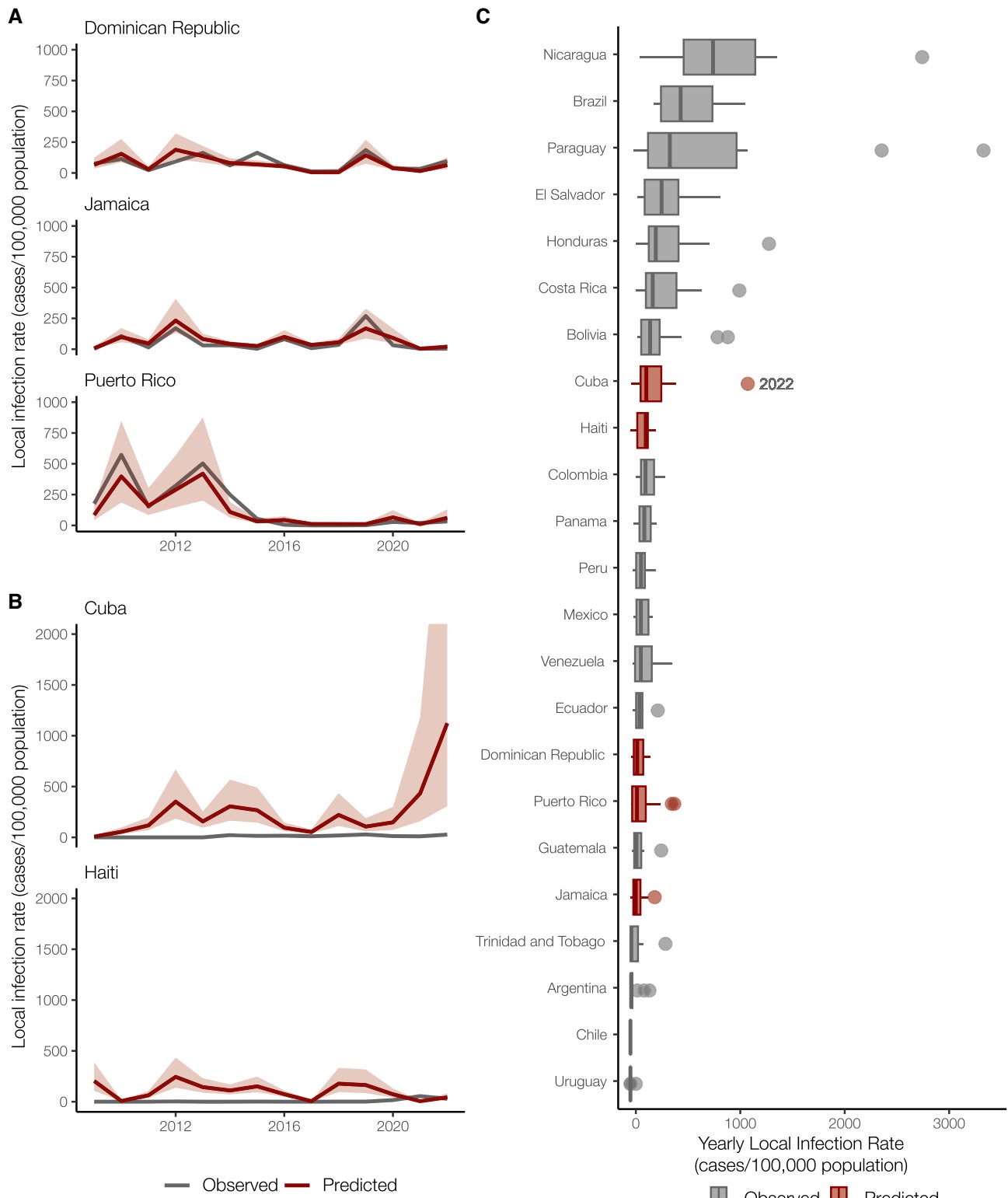

**Fig. 3 | Travel infection rates provide an estimate of local dengue infections.**
**A** Local and travel-associated dengue infection rates from the Dominican Republic, Jamaica, and Puerto Rico were used as predictors in a negative binomial regression model due to the strong correlations between local and travel-associated cases. The red lines indicate the predicted mean infection rates, and the shaded area indicates the 95% confidence interval, while the gray lines indicate the local dengue cases as reported to PAHO. **B** Local dengue infection rates in Cuba and Haiti from 2009 to 2022 were predicted using the model from panel **A**. The colored lines indicate the predicted mean infection rates, and the shaded area indicates the 95% prediction interval, while the gray lines indicate the local dengue cases as reported to PAHO. **C** Observed and predicted local infection rates for South American and Caribbean countries and territories with populations over one million inhabitants from 2009 to 2023. Within each box, thick vertical lines denote median values (50th percentile); boxes extend from the 25th to the 75th percentile; horizontal lines extending from boxes denote mark the 5th and 95th percentiles, and the dots denote outliers, representing large outbreaks from any year. Locations in gray with observed cases were not included in our model.

Haiti were not reported to PAHO for 2009–2011, 2014, and 2016–2017[7]. To estimate the number of cases that likely went under-reported in Cuba and Haiti during our study period, we constructed a negative binomial regression model to predict local infection rates from the travel rates (Fig. 3A). We found that when controlling for the gross domestic product (GDP; a relative approximation of resources available for surveillance), population, and year, the risk associated with local cases of dengue increases by 96% (95% confidence interval [CI]: 63–134%) for every one unit increase in the log of travel infection rates from the Dominican Republic, Jamaica, and Puerto Rico.

When applying our model to travel data from Cuba and Haiti, we estimate new dengue case dynamics (Fig. 3B) and compare the predicted infection rates to other observed rates throughout the Americas (Fig. 3C). We estimate that the average yearly dengue infection rates from Haiti (244 [95% CI: 138–434]/100,000 population) are similar to the reported and estimated rates from the Dominican Republic (188 [95% CI: 111–320]/100,000), which is reasonable given that they are on the same island. We also estimate that from 2010-2021, the peaks in dengue infection rates from Cuba were similar in magnitude to other Caribbean islands (-500/100,000 population). However, we predict that Cuba likely experienced a large outbreak in 2022 (-1000 [95% CI: 306–4117]/100,000 population), potentially one of the largest in the region. Our reconstructed dengue infection rates using travel surveillance provide a clearer picture of dynamics in the Caribbean and allow us to investigate previously unreported trends.

### Asynchrony of dengue virus serotypes across locations

Using travel-associated dengue cases, we reconstructed the infection trends in different locations of the Caribbean (Fig. 3). To obtain a deeper resolution of the outbreaks, we determined the DENV serotype from infected individuals with recent travel to the Caribbean and estimated the annual proportions from each location from 2010 to 2022. We discovered that outbreaks on different islands were often caused by different serotypes, even when those outbreaks occurred during the same years (Fig. 4).

While some countries report DENV serotype data associated with cases to PAHO, the data are released to the public as a binomial (present or absent) without accompanying proportions. Therefore, it is often difficult to determine (1) which serotypes are mostly responsible for outbreaks and (2) the temporal trends that may be useful for forecasting. To overcome this data limitation, we determined the serotype data for all travel-associated dengue cases from the FDOH from 2010 to 2022 (2009 serotype data were not available; Fig. 4; Supplementary Table S1). We analyzed the data as the total number of travel cases by serotype (Fig. 4A) and the proportion of each serotype (Fig. 4B) per year and country or territory. As a form of validation, our serotype proportions estimated from infected travelers returning from Puerto Rico generally match the proportions from local cases reported from Puerto Rico, including the dominance of DENV-1 from 2010 to 2013, the rise in DENV-4 from 2012 to 2014, the rise of DENV-2 from 2014 to 2015, and the re-emergence of DENV-1 in 2020[37]. Overall, we found multiple serotypes for several years and locations, which has often been reported, but years with high travel infection rates were typically dominated by a single serotype (e.g. DENV-3 from Cuba in 2022 and DENV-1 from the Dominican Republic in 2019).

As is expected for DENV, we found that different serotypes transitioned in and out of dominance in each location during the twelve years (Fig. 4). This is consistent with genotype replacement events, which occur when a previously dominant lineage is replaced by another related, but distinct, lineage[38–40]. Various theories have been proposed to explain these events, including natural selection, immune pressure, and population bottlenecks. During the earlier years, however, we did detect some patterns. We found that from 2010 to 2016, DENV-1 and DENV-2 predominated in the Dominican Republic, Haiti, Jamaica, and Puerto Rico. Immediately following the 2015-2016 Zika

epidemic, dengue cases decreased throughout the Americas[41]. As dengue cases began to increase again to record highs in 2019 (>3 million cases reported to PAHO), we found that many outbreaks were caused by different serotypes. The most prevalent serotypes in 2019 were DENV-1 from the Dominican Republic (96%, $n = 25$) and Haiti (88%, $n = 8$), DENV-2 from Cuba (90%, $n = 213$), and DENV-3 from Jamaica (87%, $n = 23$). By 2022, DENV-1 was the most commonly detected serotype in Puerto Rico (100%, $n = 6$), DENV-2 arose in the Dominican Republic (90%, $n = 10$), and DENV-3 became the new dominant serotype in Cuba (74%, $n = 735$). The only location that consistently did not follow other Caribbean serotype trends, even before the Zika epidemic, was Cuba. Overall, we demonstrate that dengue outbreaks in the Caribbean cannot be treated as a single entity, even when they are synchronized throughout the region (e.g. 2019), as transmission in different islands can be due to different DENV serotypes regardless of temporal relationships.

### Sequencing travel infections reveals dengue virus diversity in the Caribbean

Our analyses show that dengue outbreaks in the Caribbean exhibit serotype variability between islands despite temporal synchrony (Fig. 4). While serotype-level data can provide broad-scale information, there is substantial within-serotype diversity. Therefore, to uncover the DENV genetic diversity and patterns of spread within the Caribbean, we sequenced 295 traveler infections (Fig. 5).

Using sequencing data that we generated from travel-associated dengue cases, we performed four discrete phylogeographic analyses, one for each serotype (Fig. 5). The travel origin location in our analysis was designated as the site of presumed DENV exposure. We subdivided the Caribbean into Cuba, Dominican Republic, Haiti, Jamaica, Puerto Rico, and "other Caribbean". For context, we used genomic data from North America, South America, and Central America, and representative DENV sequences sorted by WHO regions for the other areas of the world to highlight viral transmission into and within the Caribbean. Our data revealed the continued circulation of all four DENV serotypes and multiple clades within those serotypes in different parts of the Caribbean as recently as 2022. While most of the sequenced travel infections cluster within clades that have historically circulated within their respective regions, we also detected several introductions of all four serotypes into the Caribbean from southeast Asia, the western Pacific, and elsewhere from the Americas stretching back to the 1970s to as recently as 2021 (described in more detail in the next section; Fig. 6).

To support our phylogenetic analyses, we performed three independent validations. One of our primary findings is the detection of a large DENV-3 (genotype III) clade from infected travelers returning from Cuba (Figs. 5 and 6A). First, we sequenced two samples from DENV-infected travelers returning from Cuba to Spain in 2022 and, second, we obtained sequencing data generated by the CDC Dengue Branch from travelers infected in Cuba in 2022, also provided by the FDOH. All of these sequences clustered in the same clade with the rest of the sequences from Cuba travelers. Third, we also sequenced samples from local dengue cases from the Dominican Republic collected in 2022 and found that these DENV-2 sequences clustered with DENV-2 sequences from Dominican Republic travel-associated dengue cases. These data suggest that our DENV genetic clustering patterns are not unique to our study population and sequencing methods, and likely represent local diversity.

### Phylogenetic patterns of emergence, spread, and transmission

Our analyses show that infected travelers can reveal DENV diversity within the Caribbean (Fig. 5). We next used our genomic data to uncover patterns of DENV introductions, spread, and transmission in the region (Fig. 6). Our intent here is not to explore every location-specific detail, but to showcase what is possible with even a relatively limited number of sequenced DENV genomes from travelers.

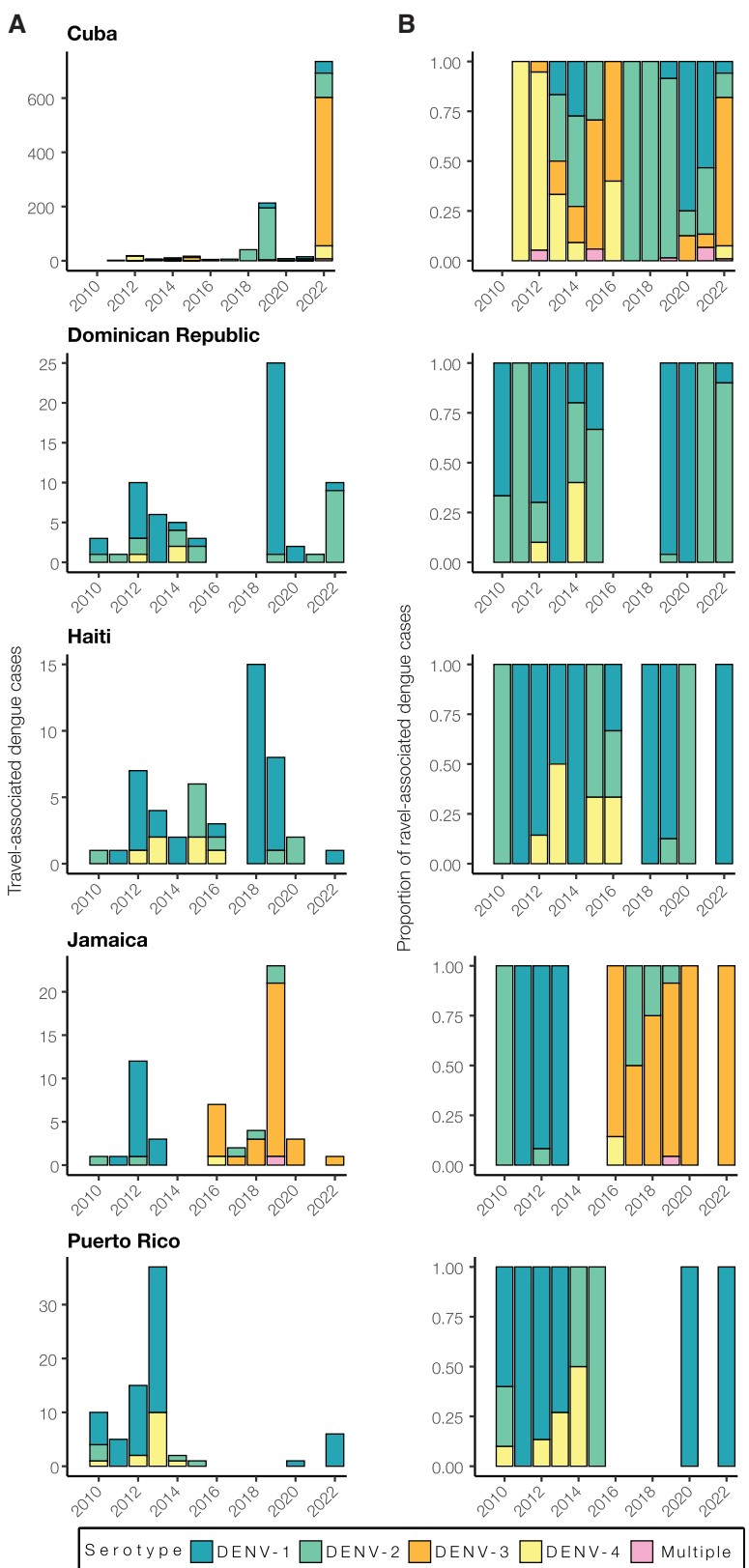

**Fig. 4 | Dengue outbreaks in the Caribbean are due to different serotypes, even during the same year. A** Yearly travel-associated dengue cases by serotype were reported by FDOH and sorted by country or territory of likely exposure. **B** The relative proportions of yearly travel-associated dengue cases by serotype per year and per country or territory of likely exposure, normalized to account for the number of infected travelers.

In our analysis, we detected a substantial number of travel-associated dengue cases coming from Cuba during 2022 (Fig. 2), and we estimated that these stemmed from a large, under-reported local outbreak (Fig. 3) predominantly caused by DENV-3, genotype III (Figs. 4 and 5). Our phylogeographic analysis further shows that the large cluster of DENV-3 sequences from Cuba in 2022 (148 Cuban sequences out of 150 total sequences in the clade, posterior support for location = 1.0) was from an introduction directly or indirectly

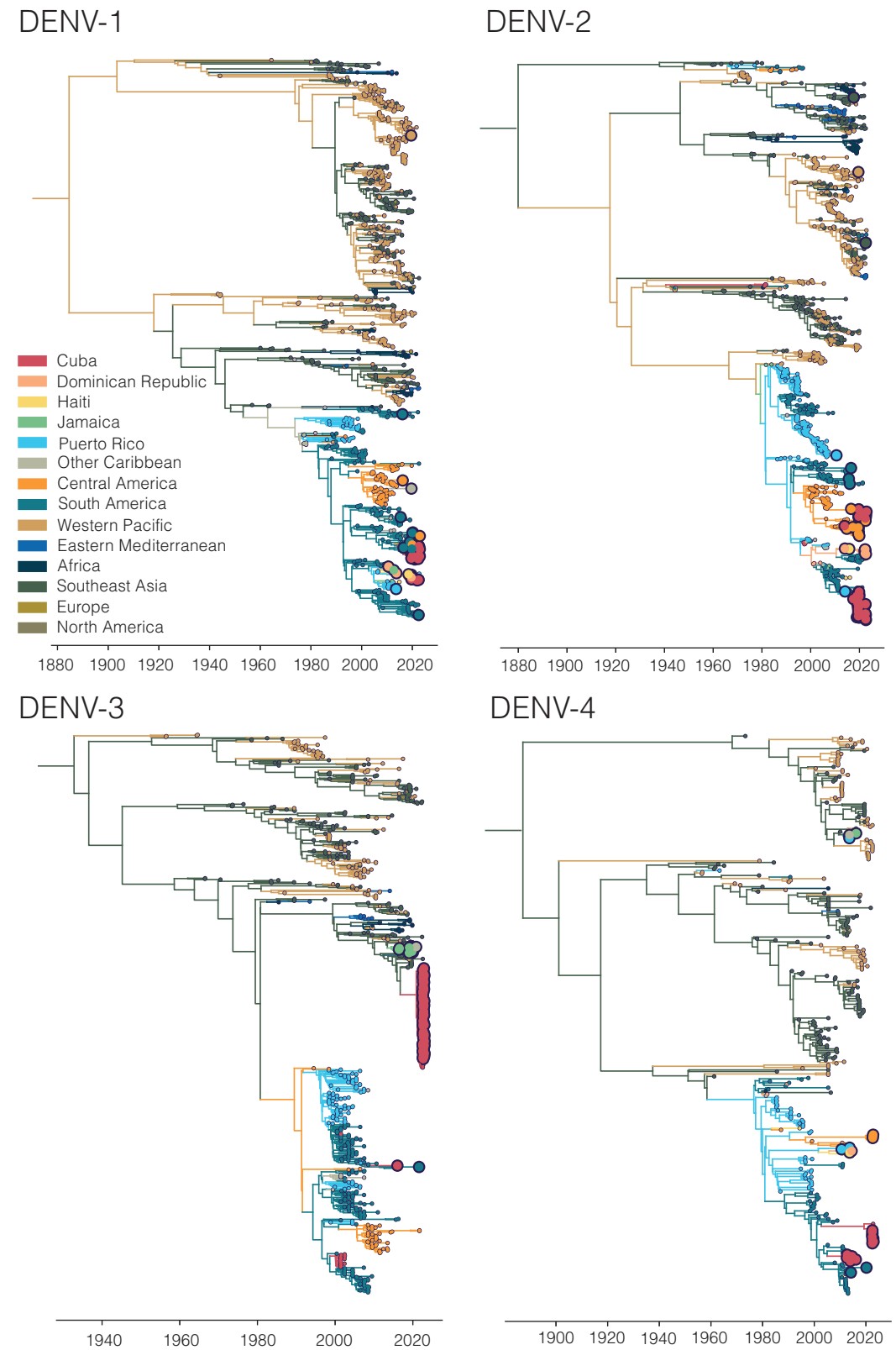

**Fig. 5 | Sequencing travel-associated dengue cases reveals genomic diversity within the Caribbean.** Time-resolved phylogenies of each serotype, with the branches and tips colored by inferred and sampled locations respectively. Larger dots represent those DENV samples sequenced for this study.

from southeast Asia that occurred by at least late 2020 (95% HPD = 2020-04-06 to 2021-05-13; Fig. 6A). We and others found that this lineage has already spread to Puerto Rico, Florida, Arizona, and Brazil by 2022[33,42]. Thus, this may be a rapidly spreading lineage of concern for the Americas.

We discovered that the 2022 Cuba outbreak clade was not the first recent reintroduction of DENV-3, genotype III, into the Caribbean from southeast Asia. Sequencing travel-associated infections also revealed an earlier but distinct introduction of this genotype into Jamaica (all 11 sequences in the clade, posterior support for location = 1.0) by at least

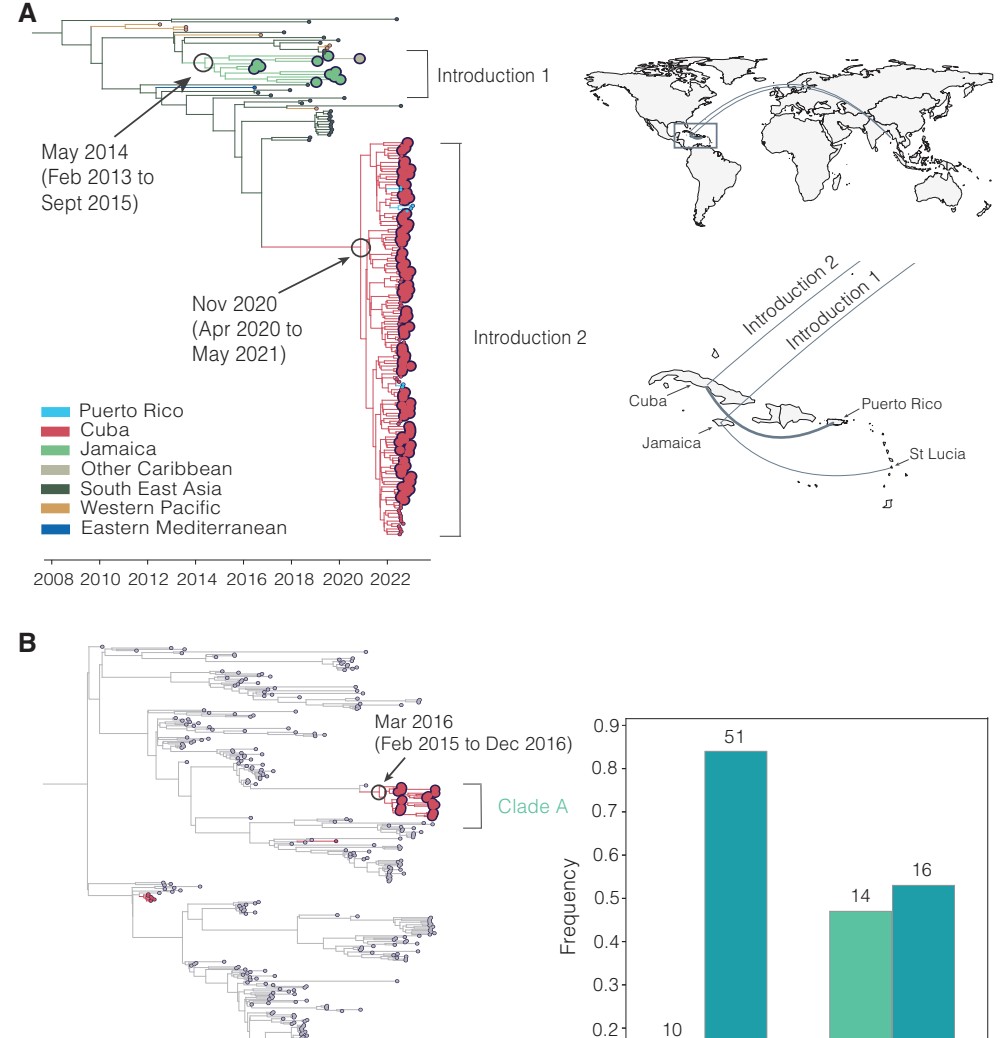

**Fig. 6 | Phylogenetic patterns of dengue virus emergence, spread, and transmission in the Caribbean. A** Time-revolved phylogeny showing global spread of DENV-3 genotype III. The times of the most common ancestor of the clades are taken conservatively as introduction times, with 95% HPDs indicated in parentheses, and are indicated by circles. The map shows the transitions from Southeast Asia to Cuba and Jamaica and within the Caribbean. Countries or territories involved are indicated on the map, and virus transitions are indicated by lines going counter-clockwise, with thickness indicating a number of movements. **B** Time-resolved phylogeny shows two co-circulating DENV-2 (genotype III) clades in Cuba, which are colored in red, with 95% HPDs indicated in parentheses. The bar chart shows the frequencies of each clade in 2019 and 2022.

2014 (95% HPD = 2013-02-02 to 2015-09-04; Fig. 6A). We first detected this lineage from three travelers returning from Jamaica in 2016 and then again from 7 travelers during the re-emergence of DENV-3 in Jamaica in 2019 (Fig. 4). We last sequenced this lineage from a traveler from St. Lucia in late 2020, indicating that the virus had limited spread, but likely not as extensively as the Cuba lineage.

In addition to lineage introductions, we also explored within-location lineage dynamics. In particular, we detected instances of multiple lineages from the same serotype co-circulating in the same country. For example, we detected two co-circulating lineages of DENV-2 (genotype III) in Cuba, designated here as clades A and B (Fig. 6B). We estimate that they were introduced into Cuba by at least 2016 (95% HPD 2015-12-20 to 2016-12-15, likely from Puerto Rico, posterior support = 0.77) and late 2017 (95% HPD 2017-02-20 to 2018-06-25, likely from Central America or Mexico, posterior support = 1.0),

respectively, and both continued to circulate through the 2019 and 2022 outbreaks. Their relative frequencies changed between 2019 and 2022, with clade B dominating in 2019 but clade A increasing to become more even in frequency in 2022, though these trends may be impacted by sampling biases. We also found evidence of co-circulating DENV-1 clades in Cuba (Fig. 5). While neither of these clades has any concerning mutational signals, evaluating the transmissibility of co-circulating lineages is the basis of discovering meaningful phenotypes that may impact the effectiveness of interventions.

## Discussion

### Travel surveillance and genomic epidemiology uncover dengue virus dynamics

Using patterns of infected travelers and virus genomics, we discovered previously unrecognized DENV dynamics within the Caribbean. Dengue

is a major public health concern within the Caribbean, and the past two decades have seen significant increases in morbidity and mortality in the region[5,34,43]. With dengue outbreaks increasing in frequency and severity[43], accurate and detailed surveillance data are crucial for epidemiological investigations. Currently, the main source of information about dengue outbreaks in the Caribbean comes from PAHO and other public health organizations, which rely on accurate case reporting from national or territorial health agencies. However, previous studies have shown large differences between case reporting to PAHO and estimated cases[34].

Dengue surveillance among travelers can be an effective method to supplement local surveillance, especially in resource-limited settings[18–20,32]. There is a large volume of travel between Florida and the Caribbean (Supplementary Fig. S2), and Florida has experienced a steady increase in imported dengue cases over the last two decades, especially from the Caribbean (Fig. 1)[44]. In this study, we investigated whether travelers with dengue returning to Florida could be used as a surrogate for local transmission in the Caribbean and found significant positive correlations between local and travel infection rates for the Dominican Republic, Jamaica, and Puerto Rico (Fig. 2B). For locations with local surveillance gaps, we built a model to estimate local infection rates in these areas, assuming the similarity between local and travel data (Fig. 3). From this analysis, we uncovered a large dengue outbreak in Cuba in 2022 (Fig. 3B) and estimated that it was similar in size to other large outbreaks detected elsewhere in the Americas (Fig. 3C). To investigate the DENV serotypes responsible for these outbreaks, we analyzed serotype data from infected travelers returning to Florida and showed that these outbreaks are not synchronized across islands, with different serotypes responsible for outbreaks on different islands within the same year (Fig. 4). We then used viral genomics to provide a snapshot of genetic diversity within the region (Fig. 5) and uncover DENV spread and introductions (Fig. 6). Overall, we demonstrate that using information about dengue in travelers, including the use of sequencing, should become more routinely used to enhance dengue surveillance.

### New DENV-3 introductions from Asia led to a large outbreak in Cuba

There have been several reintroductions of all four DENV serotypes into the Americas since its reemergence during the 1960–80s. Here, we detected two recent DENV-3 (genotype III) introductions into Cuba and Jamaica. We estimate that the introduction from southeast Asia into Cuba occurred by late 2020 (Fig. 6A) and, in the course of only two years, caused a large outbreak (Fig. 3B) and spread to several other locations in the Americas[33]. Given the speed of these events, it is possible that this DENV-3 lineage has a transmission advantage over at least some DENV lineages in the Americas.

This is not the first time that DENV-3 re-emerged in the Americas leading to widespread outbreaks. Historically, DENV-3 was re-introduced in the Americas in the 1970s, but then was not detected for 16 years between 1978 and 1994, when it was almost simultaneously isolated from Nicaragua and Panama[45–47]. It subsequently spread throughout Central America and the Caribbean, leading to widespread epidemics of dengue hemorrhagic fever[45,47–49]. DENV immune waning has previously been observed[4,50–54] and the sudden DENV-3 re-emergence stemming from that introduction could be attributed to the loss of immunity during the 16-year hiatus, which could be similarly true for the recent DENV-3 introduction into Cuba. Outside of the two new DENV-3 introductions we detected and tracked with travelers, there was very little detection of DENV-3 in the Caribbean (Fig. 3) and elsewhere in the Americas over the past 10 years[33,55,56]. While we do not yet know the true epidemiological impact of this newly introduced lineage, this will be important to closely monitor through local and travel surveillance.

Interestingly, the DENV-3 introduction into Jamaica has not spread to the same extent as the introduction into Cuba, despite its

introduction almost 6 years earlier (Fig. 6A). It could indicate that there are notable transmissibility differences between the two lineages or that the Jamaican lineage did not become established within a population with sufficient susceptibility or mobility to facilitate its spread. While these deterministic and stochastic processes require further investigation, even knowing that this introduction occurred was only possible through our surveillance among travelers - this clade was entirely sequenced from travelers (Fig. 6A).

### Genetic diversity within the region

By sequencing DENV genomes from infected Caribbean travelers, we demonstrated widespread diversity of circulating viral lineages, both between and within serotypes (Figs. 4 and 5). Even in outbreak years, such as 2019 or 2022, there was not one lineage that became dominant in the region; instead, different serotypes were often responsible for outbreaks on different islands, even if they occurred at the same time (Fig. 4). This indicates synchronized outbreaks are more likely driven by population (e.g. time since last outbreak) and environmental effects (e.g. El Niño years) than virus-related factors.

Most dengue-endemic areas in the Americas have co-circulation of multiple serotypes, which increases the likelihood of concurrent infections[57–59]. In fact, the first-ever confirmed case of concurrent DENV infection was reported in Puerto Rico during the 1982 outbreak in a patient infected with DENV-1 and -4 serotypes[58,60]. However, we noted minimal numbers of mixed infections when reviewing serotype data (14 out of 1328, Fig. 4), which is significantly lower than those previously reported[57]. Given the size of our dataset, it is possible that the overall likelihood of concurrent infections, even in hyperendemic locations, is actually quite small, as is their overall impact on severe dengue cases and epidemics.

### Limitations

Travel surveillance data is limited by location-specific patterns and volumes of travel[19,25]. During our study period, there were years in which we detected minimal or no infected travelers despite local dengue cases being reported to PAHO (Fig. 2A). This introduced biases in our local case estimates (Fig. 3) and in our patterns of DENV emergence and spread throughout the region (Fig. 6), as we have no travel data from that time period.

For this study, we assumed that the risk of DENV infection was similar for travelers and local inhabitants in endemic countries and territories, meaning that a rise in local cases would correspond to a relative increase in travel-associated infections. However, this may not be the case and would depend on the travelers' previous exposure to dengue and related flavivirus infections or vaccinations, as well as on the general patterns of travelers' behaviors while visiting the country or territories (e.g. DENV infections are more common among individuals visiting their friends and relatives than people traveling for other reasons). Endemic locations also experience heterogeneity with regard to dengue risk, as the primary vector for dengue in the Caribbean is *Aedes aegypti*, which is found more commonly in urban areas. It is also possible that some residents in endemic locations would be less likely to seek medical care and thus be diagnosed with only mild dengue symptoms, as the illness is a common occurrence for them. Thus, using travelers as sentinels alone cannot provide a complete picture of local dengue outbreaks.

Calculating travel infection rates from Cuba has important limitations we tried to address during our analysis. First, due to the associations between local dengue outbreaks in Florida and infected travelers from Cuba, the FDOH enhanced its case finding among individuals reporting recent travel from this location. The enhanced case finding was particularly apparent in 2019 and 2022, when the FDOH reported 413 and 929 travel-associated dengue cases from Cuba, respectively. We attempted to correct this by only using cases in our calculations that would have been detected in those years using the standard practice

(i.e. not using "travel to Cuba" as the reason for DENV testing since that is not common practice for travel from other locations). In addition, there was also difficulty with ascertaining an accurate number of travelers from Cuba to Florida, as not all travel between these two areas is documented, including boat travel. We do not expect this to significantly impact our travel volume estimates as we previously thoroughly investigated travel patterns from Cuba to Florida[18], and undocumented boat travel is estimated to be only 1.18% (6,182/524,611 in 2022) of the travel volumes (air passenger journeys) that we obtained from the US Department of Transportation[61]. The COVID-19 pandemic also impacted travel throughout the region (Supplementary Fig. S2) and could have impacted how people interacted with the medical system (e.g. avoiding hospitals, fevers presumed secondary to SARS-CoV-2). This makes it difficult to interpret data from 2020, 2021, and, to some extent, 2022, although for that last year, travel restrictions were mostly lifted, and we detected increased travel within the region (Supplementary Fig. S2). Thus, the potential inaccuracies in our travel infection rates from Cuba mean that our modeled local estimates should be viewed as relative rates rather than as absolute values. The compilation of our analysis still indicates that there was a large under-reported dengue outbreak in Cuba in 2022, a finding that was independently discovered from travelers returning from Cuba to other locations[32].

Utilizing infected travelers as a means of detecting local outbreaks should not replace local dengue surveillance; however, it can be helpful for locations where local surveillance is not feasible or is underfunded. By supplementing local data, travel surveillance can assist with detecting local outbreaks and estimating their size. When combined with genomic epidemiology, we used travel surveillance to provide a snapshot of the diversity of DENV within the Caribbean, as well as reveal patterns of DENV emergence and lineage spread. While we did not address it in this study, travel surveillance can also provide important epidemiological information at the destination (Florida, in this case) and could be incorporated into forecasting models. If this framework was incorporated into a routine system, utilizing networks such as GeoSentinel, over time, it would help support global surveillance efforts to combat the surge of dengue.

## Methods

### Ethics
The Institutional Review Boards (IRB) from the FDOH, Universidad Iberoamerican (UNIBE), Florida Gulf Coast University, and the Yale University Human Research Protection Program determined that pathogen genomic sequencing of de-identified remnant diagnostic samples as conducted in this study is not research involving human subjects (Yale IRB Protocol ID: 2000033281; UNIBE-CEI# 2021-88).

### Travel-associated cases, serotypes, and infection rates
Weekly cumulative reports on travel-associated dengue case numbers were collected from 2009 to 2022 and are publicly available from the FDOH[62]. These cases reported on the FDOH database include those that were confirmed by both PCR and serologic assays. Serotype data for travel-associated dengue cases was determined by the FDOH using the CDC DENV-1–4 RT-PCR Assay[63]. Travel-associated cases occurred in individuals who had traveled to a dengue-endemic country or territory in the two weeks prior to symptom onset. For this study, we only included patients who traveled to one endemic location within the 2 weeks prior to symptoms onset so we could more accurately sort the temporal and spatial distribution of travel-associated cases. This led to the exclusion of 28 patients who traveled to multiple countries during our study period, constituting 1.2% of the 2300 total cases (Supplementary Table S1). Within the Caribbean, the focus of this paper, there were 18 patients that were excluded due to traveling to multiple countries, constituting 1% of the 1815 total cases. Therefore, due to the small number of these cases, we did not feel that excluding

them would affect our analysis. We aggregated the data by year and by location of likely exposure (i.e., travel origin).

Dengue is a reportable disease in the United States. In addition to health care provider and laboratory reporting, the FDOH uses syndromic surveillance to aid in case identification. FDOH detected syndromic surveillance cases by searching de-identified chief complaints, admission and discharge diagnoses, travel-related fields, etc., obtained from participating emergency rooms and urgent care centers in Florida. Information from suspected dengue cases are forwarded to the county health departments, who in turn reach out to the local hospitals and ascertain whether dengue testing was ordered. Enhanced surveillance was increased during 2019 and 2022 due to the large number of travel-associated cases being detected. These efforts involved adding recent travel to Cuba as a criterion, in addition to dengue-like symptoms. Eighteen of the 413 travel-associated cases in 2019 and 397 of the 929 cases in 2022 were first identified via syndromic surveillance and used in our analysis. Of note, 116 cases identified via enhanced surveillance in 2022 would not have met our case definition without this additional criterion. Therefore, in order to counter these differences in case ascertainment, we only included cases captured via traditional reporting, not enhanced detection of cases among travelers returning from Cuba, for our calculations.

When we compared the local and travel case trends for Cuba with versus without the enhanced surveillance, we found that excluding these travel cases led to decreased correlation (Pearson $r = 0.575$, $p = 0.032$ versus $r = 0.496$, $p = 0.070$, respectively). When we compared the travel and local infection rates for Cuba with versus without the enhanced surveillance, we once again found that case exclusion led to decreased correlation (Pearson $r = 0.516$, $p = 0.059$ versus $r = 0.458$, $p = 0.1$, respectively). Therefore, we determined that including all the data would increase our estimates of the outbreak sizes in Cuba, and chose to use only cases captured by traditional reporting to not overestimate these local outbreaks.

Yearly travel infection rates from all exposure (origin) and reporting (destination) combinations were calculated by the number of travel-associated cases per 100,000 airline passenger journeys (from origin to destination/year). Exposure-reporting combinations that accounted for less than 80 imported cases were not included in this analysis; therefore, only data from Cuba, Dominican Republic, Haiti, Jamaica, and Puerto Rico are included. Air travel data were obtained as detailed below.

### Air passenger volumes
We collected air passenger volumes to calculate dengue travel infection rates. We obtained the number of passengers traveling by air from Cuba, the Dominican Republic, Haiti, Jamaica, and Puerto Rico that disembarked in the US port of entry in Florida from 2009 to 2022 as international and domestic non-stop segment traffic reports (T100) from the Office of Data, Analytics, and Technology (DAT), Division of Global Migration Health (DGMH). The flight travel volume data were provided by OAG Aviation Worldwide Ltd. OAG DOT Analyser, T100 International and Domestic Segment Traffic Reports, Version 2.6.1 2023 (https://analytics.oag.com/analyser-client/home; accessed May 16, 2023). United States Department of Transportation (DOT) data includes all commercial chartered and military flights arriving in airports in the United States and includes origin and destination. One limitation of DOT data is that it does not include connecting flight information. However, as this would lead to a potential overestimation of total travelers and, therefore, potentially an underestimation of travel infection rates, we felt this was reasonable.

Cruise and boat travel data were not included. There were very few infections linked with boat travel in our dataset, even between Cuba and Florida. There were also very few infections linked to cruise travel. It may be that these cases are more likely to be tourists who are subsequently diagnosed elsewhere (and just visiting Florida for the

cruise departure) or that they may have different risk behaviors compared to air travelers.

## Local cases and infection rates

PAHO is the primary source for information regarding dengue spread in the Americas, with data on suspected and confirmed cases for countries and territories within the Americas with local transmission[7]. We obtained yearly dengue case counts in Cuba, Dominican Republic, Haiti, Jamaica, and Puerto Rico from 2009 to 2022, as well as local dengue infection rates (suspected and confirmed dengue cases per 100,000 population). Infection rates were used as an estimate for infection likelihood to investigate sources of dengue introductions.

## Estimated local dengue cases

We constructed a negative binomial regression model to predict local infection rates from the travel infection rates due to evidence of overdispersion in the initial Poisson regression model that was tested using the performance package in R. Our final model can be seen below:

$$Y_t \sim Negative\,Binomial\left(r, \frac{r}{r + \lambda_t}\right), \quad (1)$$

$$\begin{aligned}\ln(\lambda_t) = \ln(pop_t) + \beta_o + \beta_1 * \log(Travel\,Infection\,Rate + 0.02) \\ + \beta_2 * GDPcat + \beta_3 * Year\end{aligned} \quad (2)$$

Where $Y_t$ is the number of cases at time period $t$, $r$ is a measure of overdispersion in the data and $\lambda_t$ is the expected number of cases at time period $t$. In the regression equation, $pop_t$ is the country population at time $t$ as an offset term for expected number cases to get an infection rate. We used two data types: locally acquired dengue cases by country and travel-associated dengue cases by country of origin to inform estimates of local infection rates in Cuba and Haiti on a scale comparable to local infections in the other three locations. A logarithmic relationship between travel and local infection rates was specified to account for high travel infections (Travel Infection Rate), particularly in Cuba. The value of 0.02 was added to avoid computational problems when travel infection rates equaled 0 and was selected via sensitivity analysis (lowest AIC score). Because we theorized that countries and territories with a higher GDP (obtained from https://worldpopulationreview.com/countries) would be able to devote more resources to finding and reporting local dengue cases, we created the variable GDPcat by income, with lower-middle as GDP <$12,535 and upper as >$12,535. This ultimately placed Puerto Rico in the upper category and stratified the other four countries in the lower category. To account for the temporal trend in the outcome, we included a Year as a linear predictor in the model. We tested for autocorrelation in Puerto Rico, Jamaica, and the Dominican Republic separately using the Durbin–Watson test and found no significant autocorrelation ($p = 0.90$, $0.91$, and $0.89$, respectively). In addition, we tested a mixed model allowing for a country- or territory-specific random GDP effect that ultimately performed worse (AIC 752 compared to 749 for the fixed model) R code and data are available at: https://github.com/grubaughlab/2023_paper_DENV-travelers

## Dengue virus sequencing

Patient serum samples were selected for study inclusion according to Ct values from prior FDOH Trioplex and serotype-specific PCR diagnostics and/or volume availability. In general, samples with a Ct value < 35 were aliquoted to provide 160 μL per sample. Lesser volume samples were included if Ct value < 25. All samples were de-identified to maintain patient privacy and blind the study.

140 μL or less of each sample was manually extracted using the QIAamp Viral RNA Mini Kit (Qiagen, Germantown, MD) following the protocol outlined in the manual unless otherwise specified. Linear

acrylamide (Invitrogen by Thermo Fisher Scientific, Waltham, MA) was substituted, 1 μL of 5 mg/mL per sample, in place of the kit provided carrier RNA. Each sample was eluted in 60 μL of ultrapure, nuclease-free water (Invitrogen by Thermo Fisher Scientific, Waltham, MA) and stored at −80 °C. Sample extracts were kept frozen on dry ice during transport and shipped to collaborators for subsequent sequencing.

Dengue virus RNA was sequenced using a pan-serotype, highly multiplexed PCR approach called 'DengueSeq' (a derivative of PrimalSeq[64]). A detailed protocol, including the DENV1-4 primer schemes, is available[65,66]. In brief, we prepared libraries using the Illumina COVIDSeq test (RUO version) with the DENV1-4 primers, and pooled libraries were sequenced on the Illumina NovaSeq (paired-end 150), targeting 1 million reads per individual library. Consensus genomes were generated at a depth of coverage of 20X and a minimum frequency threshold of 0.75 using our DengueSeq bioinformatics pipeline[67], which includes iVar[68]. All DENV genomes and sequencing data are available on BioProject PRJNA951702.

## Phylogenetic analyses

To account for differences in sampling intensity globally and to highlight dynamics within the Caribbean; we performed sample selection on the global background dataset. We used all sequences sampled from the Caribbean and then took up to 10 sequences per country or territory per year per serotype, with a strict coverage cut-off applied to all sequences of 70% of genome coverage. We then aligned these sequences with MAFFT v.7.490[69] and manually curated them using Geneious v 2022.1.1 (https://www.geneious.com). We generated an initial Maximum Likelihood (ML) tree for each serotype separately using the HKY substitution model[70] in IQTree v2.1.4[71]. We put this ML tree into TempEST[72] to assess the molecular clock signal and to identify molecular clock outliers, which were then removed[72]. Any remaining sylvatic sequences were also removed. The final dataset sizes therefore were DENV-1 $n = 1095$, DENV-2 $n = 1406$, DENV-3 $n = 839$, and DENV-4 $n = 345$.

In the interests of computational efficiency, we split the analysis into topology estimation and phylogeographic inference. We began by estimating the topology in BEAST v1.10.5[73] using a generalized time reversible (GTR) model[74], a gamma rate heterogeneity model with four categories[75], and a relaxed molecular clock[75] as in ref. [41]. The mean clock rate of each tree, using a CTMC scale prior on the mean and an exponential prior on the standard deviation, and their 95% HPDs are: $6.61 \times 10^{-4}$ ($5.81 \times 10^{-4}$ to $6.41 \times 10^{-4}$), $8.20 \times 10^{-4}$ ($7.78 \times 10^{-4}$ to $8.65 \times 10^{-4}$), $6.69 \times 10^{-4}$ ($6.28 \times 10^{-4}$ to $7.05 \times 10^{-4}$) and $7.68 \times 10^{-4}$ ($6.34 \times 10^{-4}$ to $8.94 \times 10^{-4}$) substitutions/site/year for DENV-1 to -4, respectively.

We used a Skygrid coalescent model[76] with a Hamiltonian Monte Carlo (HMC) operator[77]. We defined the gridpoints for effective population size estimation externally to be at the start of each year, every year until 2000, and then every 25 years until 1900 for DENV-1 and DENV-3, and every 50 years until 1710 for DENV-2 and every 50 years until 1750 for DENV-4. These gridpoints aimed to balance resolution in more recent years against computational efficiency, and final dates were placed slightly before the estimated roots of the trees, as described in[78].

The length and number of chains required for convergence and sufficient ESS values depended on the serotype but ranged between two and four chains of 100–600 m states with 10–60% removed for burn-in.

From each of these analyses, we randomly sampled 500 trees from the post-burn-in posterior distribution to use as an empirical tree distribution to infer the geographic spread of each dengue virus serotype. We grouped together locations of sequences, starting with WHO regions (Africa, Europe, eastern Mediterranean, Southeast Asia, and western Pacific). The Americas were further split into South America, Central America plus Mexico, and the rest of North America. We also split the Caribbean into Jamaica, Haiti, Cuba, Dominican

Republic, Puerto Rico, and other Caribbean. In total, there were 14 regions for DENV-1 and DENV-2, 12 for DENV-4, and 10 for DENV-3. Details of the number of sequences for each region can be found in Supplementary Table S1. We performed a discrete asymmetric phylogeographic analysis using these regions and a CTMC prior[79]. Each serotype analysis was run for two chains of 10 m states, each with 10% removed for burn-in. Assessment of convergence for every analysis was performed using Tracer[78].

Introductions were inferred using a custom Python script. We defined an introduction as the oldest node on a branch which transitioned into the territory/region of interest and counted all downstream nodes as part of the same introduction regardless of further exports/reintroductions. The time of introduction was taken conservatively, as the time of the transition node itself, and should be interpreted as the latest approximate time that the introduction could have occurred (i.e. the introduction could have occurred earlier, but not later). The XML files are available at: https://github.com/grubaughlab/2023_paper_DENV-travelers.

### Reporting summary

Further information on research design is available in the Nature Portfolio Reporting Summary linked to this article.

## Data availability

All DENV genomes and sequencing data generated in this study are available on BioProject PRJNA951702. Alignments provided in our data repository include all GenBank accession numbers for any publicly available sequences that were analyzed in this study (https://github.com/grubaughlab/2023_paper_DENV-travelers). The travel surveillance data is available in the Supplementary Information file. The air passenger data used in this study are proprietary and were purchased from OAG Aviation Worldwide Ltd. These data were used under the United States Centers for Disease Control and Prevention license for the current study, and so are not publicly available. The authors are available to share the air passenger data upon request and with the permission of OAG Aviation Worldwide Ltd.

## Code availability

All code and model results are available at: https://github.com/grubaughlab/2023_paper_DENV-travelers.

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

## Acknowledgements

We acknowledge S. Taylor and P. Jack for technical discussions. This publication was made possible by the National Institute of Allergy and Infectious Diseases of the National Institutes of Health (NIH) under Award Number DP2AI176740 (NDG), CTSA Grant Number UL1 TR001863 from the National Center for Advancing Translational Science, a component of the NIH (CBFV), the NIH T32AI055403 (AS), National Science Foundation (NSF) Graduate Research Fellowship under Grant No. DGE-2139841 (AS), The Dominican Ministry of Higher Education and Technology Fund (FONDOCYT-2022-1B3-069), the National Institute of General Medical Sciences R21GM142011 (SFM), and through a Cooperative Agreement between the US Centers for Disease Control and Prevention (CDC) and the International Society of Travel Medicine (ISTM) (Federal Award Number: 1 U01CK000632-01-00). The findings and conclusions in this report are those of the author(s) and do not necessarily represent the official position of the CDC, NIH, or FDOH.

## Author contributions

E.T.S, C.B.F.V., A.M., S.F.M., and N.D.G. designed the study. E.T.S., V.H., L.M.P., and R.T.K. wrote the paper. L.M.P., A.M., S.Bu., N.C., M.C., S.E., A.G., N.G., A.H., L.H., R.J., E.K., P.C., A.D., S.Ba., E.R., A.M.E., B.S., D.S., R.Z., R.P.R, S.M., A.P., N.S., F.F.N., J.C.G., J.L.M.J., G.A.S, L.E.A, G.P.B., R.H., and D.H.H. collected clinical samples and data. M.I.B, C.C., A.S., V.H., and C.B.F.V. performed the sequencing and bioinformatic analysis. M.S. and V.K. provided travel data. E.T.S., R.T.K., V.H., and J.L.W. performed statistical and phylogenetic analyses. N.D.G, C.B.F.V, A.M. and S.F.M supervised the project. All authors reviewed and approved the final version of the manuscript.

## Competing interests

N.D.G. is a paid consultant for BioNTech for work unrelated to this manuscript. All other authors declare no competing interests.

## Additional information

¹Department of Pediatrics, Yale School of Medicine, New Haven, CT, USA. ²Department of Epidemiology of Microbial Diseases, Yale School of Public Health, New Haven, CT, USA. ³Department of Biological Sciences, College of Arts and Sciences, Florida Gulf Coast University, Fort Myers, FL, USA. ⁴Department of Biostatistics, Yale School of Public Health, New Haven, CT, USA. ⁵Public Health Modeling Unit, Yale School of Public Health, New Haven, CT, USA. ⁶Bureau of Public Health Laboratories, Division of Disease Control and Health Protection, Florida Department of Health, Tampa, FL, USA. ⁷Bureau of Public Health Laboratories, Division of Disease Control and Health Protection, Florida Department of Health, Jacksonville, FL, USA. ⁸Florida Department of Health in Miami-Dade County, Miami, FL, USA. ⁹Bureau of Epidemiology, Division of Disease Control and Health Protection, Florida Department of Health, Tallahassee, FL, USA. ¹⁰Division of Vector-Borne Diseases, Centers for Disease Control and Prevention, San Juan, Puerto Rico. ¹¹Office of Data, Analytics, and Technology, Division of Global Migration Health, Centers for Disease Control and Prevention, Atlanta, GA, USA. ¹²Bureau for Global Health, United States Agency for International Development, Arlington, VA, USA. ¹³Instituto de Medicina Tropical & Salud Global, Universidad Iberoamericana, UNIBE Research Hub, Santo Domingo, Dominican Republic. ¹⁴National Referral Unit for Tropical Diseases, Infectious Diseases Department, CIBER de Enfermedades Infecciosas, IRYCIS, Hospital Ramón y Cajal, Universidad de Alcalá, Madrid, Spain. ¹⁵Microbiology Department, Hospital Ramón y Cajal, Instituto Ramón y Cajal de Investigación Sanitaria (IRYCIS), CIBER de Epidemiologia y Salud Publica (CIBERESP), Madrid, Spain. ¹⁶Department of Infectious Tropical Diseases and Microbiology, IRCCS Sacro Cuore Don Calabria Hospital, Negrar, Verona, Italy. ¹⁷Department of Global Health, Boston University School of Public Health, Section of Infectious Diseases, Boston University School of Medicine, Center for Emerging Infectious Disease Policy and Research, Boston University, and National Emerging Infectious Disease Laboratory, Boston, MA, USA. ¹⁸Yale Institute for Global Health, Yale University, New Haven, CT, USA. ¹⁹Department of Ecology and Evolutionary Biology, Yale University, New Haven, CT, USA. ²⁰These authors contributed equally: Emma Taylor-Salmon, Verity Hill, Lauren M. Paul, Robert T. Koch. ²¹These authors jointly supervised this work: Chantal B. F. Vogels, Andrea Morrison, Scott F. Michael, Nathan D. Grubaugh. ✉e-mail: emma.taylor-salmon@yale.edu; andrea.Morrison@flhealth.gov; smichael@fgcu.edu; nathan.grubaugh@yale.edu

