## [Peer Review File · Nature Communications]

Travel surveillance uncovers dengue virus dynamics and introductions in the CaribbeanREVIEWER COMMENTS

Reviewer #1 (Remarks to the Author):

In this manuscript, the authors estimate DENV caseloads on different Caribbean islands, with different local surveillance from traveler cases to Florida. To do so, they model the observed number of positive travelers returning from the different islands as negative binomial distributions from the then-estimated local caseloads. Additionally, the authors sequence DENV from travelers to reconstruct the DENV serotype dynamics. Using these approaches, they then reconstruct, in particular, a recent outbreak of DENV in Cuba.

The manuscript is a good and interesting paper that reads very well. My main points of criticism of the manuscript as is, are:

For most locations, the traveler corresponds to local caseloads, except for Cuba and Haiti. I follow the reasoning for why Haiti would not correspond, though I don't understand why there is a spike in PAHO data in 2021 but not traveler data. From looking at Figure 2, Cuba seems to have consistent reporting, yet the traveler data do not really match; why? Also, is there any direct evidence for the large Cuba outbreak? The CIs are huge and include a massive but also no exceptional outbreak (Fig 3 B).

I would downplay some of the phylogeographic findings. Namely, everything about the timings of introductions based on clade mrca's. Also, the substantial amount of sampling bias in the data, in my opinion, means that the number of introduction events should be treated very cautiously. I would expect the number of introduction events between two locations to be almost linearly correlated with the number of samples. As such, I'd expect these patterns to be almost entirely driven by sampling. Additionally, sampling bias here is not necessarily counteracted by using the same number of samples per location.

Minor points

L90+: confusing sentence, rephrase

L103: Therefore, we hypothesize that we can use

L112: Introduce PAHO

L138: We hypothesize that we can address this variability by...

L218+: the model should be introduced

L222+: Vase on how many observations is the statement about GDP done? Effectively 5?

L258: Confusing sentence

L284: We found that outbreaks were caused by different stereotypes.

L290: Why is Cuba considered different and e.g. not Haiti, or Jamaica?

L380: Our analyses show that there is DENV diversity within the Caribbean.

L382: This makes it sound like this is not expected or unknown. As traveler data is effectively local infection data, a limited amount of sequenced traveler data should not give you more info as the same amount of data collected locally.

L390: 2020 is the time of the mrca of the clade, not of the introduction, also, direct introduction can not be shown. The mrca will always be later than the introduction

L425: What additional a information about serotypes is provided by the phylogenetic analyses that is not already available from the sequence data alone.

L469: There isn't really an actual validation of this approach using, e.g. a simulation study

L490+: The transmission advantage statements feel very strong, considering there is a lot of variability in serotypes over time.

L569: about 1.18%, remove the about or change the 1.18%

L726: How is this accounting for sampling bias. My understanding of sampling bias is that if there is a probability of an infection being detected in location a being different than location b, then sampling is biased. This doesn't seem to correct for that.

Reviewer #2 (Remarks to the Author):

It is important to understand how dengue lineage persistence and spread in the Caribbean is driving new outbreaks in the Americas. The authors analyze the spread of dengue lineages in the Caribbean through analysis of locally reported cases, mobility data, and 295 novel dengue genomes obtained from travel surveillance cases. This study uncovered previously undetected dengue outbreaks and identified recent dengue introductions from Asia to Jamaica and Cuba. This includes a novel introduction of DENV-3 genotype III in Cuba that has spread widely in the Americas. Overall, this is a methodologically robust and sound work that significantly contributes with several advances to the field of dengue genomic epidemiology. Importantly, this study highlights the critical role of travel surveillance in complementing local surveillance. I only have minor comments.

Comments:

Page 6, line 177, Figure 2A. "Notably, we found peaks of travel-associated infections from Cuba in 2022 that were not captured by the dengue case data reported by PAHO (Figure 2A)." This is not clear from Figure 2A as there is an overlap between travel cases (red) and local dengue cases (grey).

Page 9, line 278. "As is expected for DENV, we found that different serotypes transitioned in and out of dominance in each location during the twelve years". It would be useful to include references from previous studies supporting the stated 12-year pattern for serotype replacement dynamics in the Americas.

Page 13, lines 394 and 400. It would be useful to mention how large the Cuban and Jamaica DENV3-III clades are (e.g. number of sequences from Cuba and Jamaica in relation to total number of sequences belonging to each clade).

Page 13, lines 413 and 414. Please include 95% Bayesian credible intervals for the number of virus transitions obtained from phylogeographic analyses.

Page 14, lines 421-423. The sink-source results obtained using phylogeographic analyses are sensitive to sampling biases (heterogeneity in genomic sequencing data availability across locations). I would suggest including a supplementary figure showing how representative are sequence counts in relation to the estimated number of infections in key considered locations.

Page 19, lines 561-563: "We attempted to correct this by using numbers in our calculations by only using cases that would have been detected in those years using the standard practice". Fix typo.

Page 22, lines 677 and 678. Please check the formulas of the final model.

Page 23, line 730. "70%" of genome coverage?

Reviewer #3 (Remarks to the Author):

The paper is relatively well written, has a clear and substantive narrative and is of public health importance. I do, however, have some questions and recommended additional analyses. Mainly, I found the section on the phylogeographic analyses hard to parse – particularly what constitutes an introduction, why it was defined in that manner to exclude subtree structure below the internal node of interest and why the authors did not instead opt for a Markov Jump analysis to capture the uncertainty and timing of transitions (it is unclear if they did – I spotted a logger in their XML). See below as to my thoughts on this:

Major:

222: Puerto Rico is the only country classified in the upper category, with the other four in the lower category? Not sure this is powered for interpretation, and at least needs some caveating in text for the reader to contextualize the results.

226-237: Do these countries have similar abundances of viable vectors? Do they have similar mosquito control programs? How have the control programs changed in recent times and how do you expect this to influence incidence?

278: I'm assuming there is not high resolution spatial information on what serotype circulates where, and how that is reflected in travelers?

282-83: What is this attributable to?

Line 350 onwards: Please see comments on Markov Jump analyses and clarity required for definition of an introduction below.

356-367: I'm slightly confused as to why this is framed as validation; Surely line 365-367 is a given?

391: What is the posterior support for that state? And for Jamaica on line 400.

391: This is about a year's worth of uncertainty – worth annotating the distribution around the node in the figure?

411: Again, I'm not sure if these represent median Markov jumps across time (logger in XML?), or introductions as defined in Methods section, which I find unclear and am missing the custom python script from the github. Also, these have no credible (e.g. Markov jump) or confidence intervals – phylogenetic uncertainty should be accounted for - especially as you only include 500 trees - for or are all trees highly supported?

411-: What are the timings of these introductions? Any interesting patterns in seasonality in source-sink?

418: As these numbers are fundamentally subject to sampling biases as mentioned in 420, it would be good to have an idea of how many sequences from each location are in each serotype tree. Also worth discussing how you think your downsampling framework of 10/country/year per serotype would affect this (as it's as arbitrary as all downsampling schemes!), especially as you include all Cuban sequences.

429: Need HPDs on introduction estimates – worth including in figure 6 A, C too.

430: Need posterior support values for the source locations (“likely from Puerto Rico”)

431: Do you think their relative frequencies reflect sampling biases or causality? What is the spatial resolution of this co-circulation?

491: Too strong and causal language – not proved or quantified.

499: Is there a reference/context for immune waning of DENV for the reader?

512: What do you mean by susceptibility here? Do we think these lineages are limited in subpopulations with different levels of immunity? Do you have spatial data for these sequences within Jamaica?

521: Might need expanding on what you mean by “population and environmental effects”

609: How many patients had traveled to more than one location? How do you expect this exclusion

to affect your estimates?

677-678: The notation is not parseable in the equation unfortunately for my review. Additionally, there are several parameters that require explanation in text.

694: Why only those countries?

742: Confusing phrasing – are these clock rates? Also worth mentioning the priors on the clock.

754: Were grid points subject to sensitivity testing?

758: 60% burn-in is extremely high for a chain – what were the ESS for these chains, and if they were extended e.g. doubled, were they still at equilibrium? Did the results for that dataset replicate over independent chains? 60% would suggest that a few underlying parameters were fluctuating for the majority of the chain and may indicate issues with the model.

760: Were the 500 trees sampled from the converged posterior at random? I'm assuming this was for computational tractability, but it is rather low – need to account for phylogenetic uncertainty in analyses.

773: Why did the authors not log Markov jumps across the tree, which can capture location transitions and transition times far more accurately/easily – and is parsable by the TreeMarkovAnalyzer in the Beast tools suite. In one of the XMLs examined, there seems to be a Markov jump logger – why were introductions defined as per 773-777? For example “The time of introduction was taken conservatively, as the time of the transition node itself.” – no need for this as Markov jumps include their placement in time along the branch?

773-777: I find this text very confusing – I'm not entirely sure what constitutes an introduction based on this. So, an introduction is only the first node in a lineage after a transition, with all subtending nodes assigned the same state? What if that state is poorly supported? What if there's true mixing in the lineage below it, which will be lost by assigning it the state of a node several nodes up? Again, phylogenetic uncertainty can be accounted for in Markov jump analyses across a large enough set of trees from the posterior.

773: Also I may be looking past it, but I do not see the custom python script in the github, so I can't figure out how introductions were parsed.

Minor:

167 + 173: What does the inclusion of enhanced syndromic surveillance case numbers do to your estimates for Cuba?

179: Might be worth discussing, in context of the rest of the world, why Florida in particular is a good sentinel for Cuba. It appears in the Discussion around 467, but should be set-up in the Introduction.

198: Figure 2 “travel volume to Florida from 2020-2020” – assuming the end date is a typo?

392: Are these sequences in Figure 6A?

477: Perhaps a bit overstated – not “factors”, just serotype.

519: “not one lineage”?

623: Might be a phrasing problem but 397/929 cases in 2022 does not warrant the applied “Only”

625: “Without this additional intervention” – assuming it refers to the addition of travel history to Cuba, but might want to rephrase for clarity.

Reviewer 1

In this manuscript, the authors estimate DENV caseloads on different Caribbean islands, with different local surveillance from traveler cases to Florida. To do so, they model the observed number of positive travelers returning from the different islands as negative binomial distributions from the then-estimated local caseloads. Additionally, the authors sequence DENV from travelers to reconstruct the DENV serotype dynamics. Using these approaches, they then reconstruct, in particular, a recent outbreak of DENV in Cuba.

We thank the reviewer for their positive response and for the time to review our manuscript.

For most locations, the traveler corresponds to local caseloads, except for Cuba and Haiti. I follow the reasoning for why Haiti would not correspond, though I don't understand why there is a spike in PAHO data in 2021 but not traveler data. From looking at Figure 2, Cuba seems to have consistent reporting, yet the traveler data do not really match; why? Also, is there any direct evidence for the large Cuba outbreak? The CIs are huge and include a massive but also no exceptional outbreak (Fig 3 B).

With regards to Haiti, there is very inconsistent reporting to PAHO for that country. During our study period, there were six years (2009, 2010, 2011, 2014, 2016, 2017) in which dengue cases were not reported. In some of those years (2014, 2016, 2017), not only were dengue cases not reported, but the overall population of the country was not reported. Therefore, it is difficult to say if the total case numbers in 2021 truly constitutes a spike in cases, or simply improving surveillance measures during recovery after natural disasters. There were 93 reported cases in 2019, up to 1,783 cases in 2020, 6,298 cases in 2021 and 3,316 cases in 2022. Yet in 2023 (which was not included in this manuscript), we again have a year without population or case data reported.

With regards to Cuba, while they reported no cases from 2010-2013, there has been consistent reporting to PAHO since 2014. However, there is concern that there may be under-detection/under-reporting. While we are careful not to speculate in this manuscript, this is something that has been seen before from Cuba, where they can downplay total case numbers (Grubaugh ND, *et al.* Travel Surveillance and Genomics Uncover a Hidden Zika Outbreak during the Waning Epidemic. *Cell.* 2019;178(5):1057-1071.e11). Even OpenDengue has consistent gaps in the numbers they report for Cuba (<https://opendengue.org/>).

There is no direct evidence of a large Cuba outbreak in 2022 from the country itself. However, an increased number of infected travelers returning from Cuba was seen not just in Florida, but in other areas of the world (e.g. Spain), which is inconsistent with what was reported by Cuba. And while the CIs in our model are large, the local case number reported by Cuba does not fall within them.

I would downplay some of the phylogeographic findings. Namely, everything about the timings of introductions based on clade mrca's. Also, the substantial amount of sampling bias in the data, in my opinion, means that the number of introduction events should be treated very cautiously. I would expect the number of introduction events between two locations to be almost linearly correlated with the number of samples. As such, I'd expect

these patterns to be almost entirely driven by sampling. Additionally, sampling bias here is not necessarily counteracted by using the same number of samples per location.

We agree that some of the phylogeographic results should be downplayed and that the introductions should be better explained, as also highlighted by other reviewers. To start, **we removed the section on within-Caribbean spread shown in the previous Figure 6B**. We agree that there is a substantial amount of sampling bias based on travel patterns to Florida (rather than specific local transmission intensity) that would be difficult to correct for without proportional downsampling. Our intent for that section was more to showcase how our sequencing of travelers can reveal other insights into DENV in the Caribbean and the results were not central to our overall objectives. This is substantiated by its lack of mention in the Discussion section. In retrospect, we should not have included this analysis in the final version of the manuscript and we are thus removing it.

We agree that we need to clarify the timing of the introductions, but we disagree that we should disregard using the clade MRCAs all together to discuss them. Using the MRCAs to discuss the introductions is a more conservative estimate than estimating that the introduction could have occurred anytime along the branch - suggesting potentially long-term persistence without detection. The time to the MCRA (“transition node”) and the true introduction time only diverge when there is substantial missing genetic diversity that would push the true MRCA further back in time. This is often a concern with very small datasets and/or when only a small proportion of the location is sampled (e.g. only one state in Brazil). Here we are dealing with relatively small islands with decent sampling and our tMRCA estimates have remained consistent as new data have been added since writing this manuscript. **We did make clarifications in the Results and Methods to indicate that our estimated introductions times are the latest in which they would have likely occurred.**

Results (example of change, made similar changes to other sections referencing to estimated introduction times): *“Our phylogeographic analysis further shows that the large cluster of DENV-3 sequences from Cuba in 2022 was from an introduction **directly or indirectly** from southeast Asia that ~~likely~~ occurred **by at least** in late 2020 (95% HPD = 2020-04-06 to 2021-05-13; Figure 6A).”*

Methods: *“We defined an introduction as the oldest node on a branch which transitioned into the territory/region of interest, and counted all downstream nodes as part of the same introduction regardless of further exports/reintroductions. ~~Introductions were allowed to leave the country or territory/region of interest: that is, all downstream nodes in the country or territory of the first transition node were assigned to the same introduction regardless of nodes outside of the region in between. The time of introduction was taken conservatively, as the time of the transition node itself, and should be interpreted as the latest approximate time that the introduction could have occurred (i.e. the introduction could have occurred earlier, but not later).~~”*

L90+: confusing sentence, rephrase

We **rephrased** the referenced sentence:

*“Therefore, **the areas** where ~~we need~~ disease surveillance **is** the most **needed** - like the Caribbean - are often **the same areas** where we **have** ~~with~~ limited publicly available data.”*

L103: Therefore, we hypothesize that we can use

We **edited** the referenced sentence as suggested:

*“Therefore, we hypothesized that we **can** ~~could~~ use surveillance of dengue-infected travelers diagnosed in Florida who recently returned from the Caribbean to better reconstruct DENV dynamics in the region.”*

L112: Introduce PAHO

We **clarified** the referenced sentence as suggested:

*“An increase in infected travelers returning from Cuba in 2022 has previously been reported ³², leading to the assumption of a large dengue outbreak that was not reported to the **Pan-American Health Organization (PAHO), the primary source for information regarding dengue spread in the Americas** ⁷.”*

L138: We hypothesize that we can address this variability by...

We **edited** the referenced sentence as suggested:

*“**We hypothesize that we can address this variability by using** ~~To address this variability, we hypothesized that dengue surveillance of infected travelers in Florida could be used to detect surveillance gaps within these five Caribbean islands.~~”*

L218+: the model should be introduced

The model was introduced in the referenced sentence. See below.

“To estimate the number of cases that likely went under-reported in Cuba and Haiti during our study period, we constructed a negative binomial regression model to predict local infection rates from the travel rates (Figure 3A).”

L222+: Vase on how many observations is the statement about GDP done? Effectively 5?

This observation was based on 3 observations (for Puerto Rico, Dominican Republic and Jamaica). Due to the potential impact of the small sample size brought up by the reviewers and that this part is not central to our story, we **removed the section** on the impact of GDP on dengue case reporting.

“~~We also estimate that having a higher GDP is associated with a 111% more reported dengue cases (95% CI: 12% - 314%), which supports that increased resources are associated with increased case-finding efforts.~~”

L258: Confusing sentence

We **clarified** the referenced sentence:

*“We discovered that outbreaks on different islands were often caused by different serotypes, even **when those outbreaks occurred during the same years (Figure 4).**”*

L284: We found that outbreaks were caused by different stereotypes.

We **clarified** the referenced sentence:

*“As dengue cases began to increase again to record highs in 2019 (>3 million cases reported to PAHO), we found that **many** ~~the~~ outbreaks were ~~mostly~~ caused by different serotypes.”*

L290: Why is Cuba considered different and e.g. not Haiti, or Jamaica?

Honestly, we wish we knew the answer to this question. It is something that we wish to follow up on with more broad sampling to determine how unique Cuba is with dengue virus trends.

L380: Our analyses show that there is DENV diversity within the Caribbean.

We **edited** the referenced sentence as suggested:

*“**Our analyses show that infected travelers** ~~We showed that sequencing travel-associated dengue cases can reveal DENV genetic diversity within the Caribbean (Figure 5), which is an important context to support local genomic surveillance.~~”*

L382: This makes it sound like this is not expected or unknown. As traveler data is effectively local infection data, a limited amount of sequenced traveler data should not give you more info as the same amount of data collected locally.

In countries with robust local surveillance and sequencing infrastructure, then the reviewer is correct, in that traveler data would not provide more information than locally collected data. However, in most Caribbean countries, there is a lack of local sequencing data. In the absence of sequencing data from these countries, sequencing of infected travelers provides a snapshot of what is occurring locally. This has previously been documented as an effective strategy (see references below), but has not been done with DENV in the Caribbean.

1. Grubaugh ND, *et al.* Travel Surveillance and Genomics Uncover a Hidden Zika Outbreak during the Waning Epidemic. *Cell*. 2019;178(5):1057-1071.e11.
2. Wilder-Smith A, *et al.* DengueTools: innovative tools and strategies for the surveillance and control of dengue. *Glob Health Action*. 2012;5:10.3402/gha.v5i0.17273.
3. Leder K, *et al.* GeoSentinel surveillance of illness in returned travelers, 2007-2011. *Ann Intern Med*. 2013;158(6):456-468.
4. Harvey K, *et al.* Surveillance for travel-related disease--GeoSentinel Surveillance System, United States, 1997-2011. *MMWR Surveill Summ*. 2013;62:1-23.

L390: 2020 is the time of the mrca of the clade, not of the introduction, also, direct introduction can not be shown. The mrca will always be later than the introduction

See our response to the point above about introductions. We **clarified that the times are the latest in which the introduction likely occurred**. The origin of the introduction was likely from southeast Asia, as shown in our phylogeographic analysis. Whether it came directly from southeast Asia to Cuba, though that is the most likely scenario with the available data, is not the most important part of that sentence. The key is that this is another example of an Asia to Americas introduction, as discussed further in the Discussion.

*“Our phylogeographic analysis further shows that the large cluster of DENV-3 sequences from Cuba in 2022 was from an introduction **directly or indirectly** from southeast Asia that ~~likely~~ occurred **by at least** ~~in~~ late 2020 (95% HPD = 2020-04-06 to 2021-05-13; Figure 6A).”*

L425: What additional information about serotypes is provided by the phylogenetic analyses that is not already available from the sequence data alone.

We are not exactly clear what the reviewer is asking here but we assume that it about serotype/genotype classifications versus phylogenetically defined lineages. In this case, the “sequence data alone” would classify the sequences as all the same but the phylogenetics show that there were actually two separate introductions and different transmission dynamics between the lineages.

L469: There isn't really an actual validation of this approach using, e.g. a simulation study

While we did not perform a simulation study in this paper, we were able to obtain data from locations which we know have good local surveillance (Puerto Rico, Dominican Republic, and Jamaica) that provided support for our approach.

L490+: The transmission advantage statements feel very strong, considering there is a lot of variability in serotypes over time.

We **edited** the referenced sentence:

*“Given the speed of these events, it is **possible** ~~likely~~ that this DENV-3 lineage has a ~~significant~~ transmission advantage over at least some DENV lineages in the Americas.”*

L569: about 1.18%, remove the about or change the 1.18%

We **edited** the referenced sentence:

“We do not expect this to significantly impact our travel volume estimates as we previously thoroughly investigated travel patterns from Cuba to Florida ¹⁸, and undocumented boat travel is estimated to be only ~~about~~ 1.18% (6,182/524,611 in 2022) of the travel volumes (air passenger journeys) that we obtained from the US Department of Transportation ⁵².”

L726: How is this accounting for sampling bias. My understanding of sampling bias is that if there is a probability of an infection being detected in location a being different than location b, then sampling is biased. This doesn't seem to correct for that.

Please note that we did not use the term "sampling bias" and did not intend for our methods to address this. We needed to subsample the global dataset to (1) increase the computational efficiency and (2) distribute the sequences across different countries. For the latter part, this was specifically done so that our data were not overrepresented by a few locations in Asia with the majority of the sequencing data (e.g Thailand, China). While we do not address undersampling specifically by using this method, by reducing the level of over-representation from some locations, we aim to balance the dataset and therefore reduce the overestimation of introductions from these regions.

Reviewer 2

It is important to understand how dengue lineage persistence and spread in the Caribbean is driving new outbreaks in the Americas. The authors analyze the spread of dengue lineages in the Caribbean through analysis of locally reported cases, mobility data, and 295 novel dengue genomes obtained from travel surveillance cases. This study uncovered previously undetected dengue outbreaks and identified recent dengue introductions from Asia to Jamaica and Cuba. This includes a novel introduction of DENV-3 genotype III in Cuba that has spread widely in the Americas. Overall, this is a methodologically robust and sound work that significantly contributes with several advances to the field of dengue genomic epidemiology. Importantly, this study highlights the critical role of travel surveillance in complementing local surveillance. I only have minor comments.

We thank the reviewer for their kind words and for the time to review our manuscript.

Page 6. line 177, Figure 2A. "Notably, we found peaks of travel-associated infections from Cuba in 2022 that were not captured by the dengue case data reported by PAHO (Figure 2A)." This is not clear from Figure 2A as there is an overlap between travel cases (red) and local dengue cases (grey).

While there is overlap of the two lines in Figure 2A, these graphs were scaled to peaks for both data sets. When we compare the relative proportions between years in Figure 2B, we see that there is a similar magnitude of local cases in Cuba in 2022 (grey) as was seen previously (such as in 2014 and 2019). However, there is a large increase in travel cases reported in 2022 (red line) compared to prior years.

Page 9, line 278. "As is expected for DENV, we found that different serotypes transitioned in and out of dominance in each location during the twelve years". It would be useful to include references from previous studies supporting the stated 12-year pattern for serotype replacement dynamics in the Americas.

There is not necessarily a 12-year pattern of serotype replacement within the Americas, simply that our study period was limited to those 12 years. However, there has previously been evidence of serotype fluctuation within a region (see references below), and we were therefore unsurprised to find similar trends during our study period. **We therefore updated the Discussion to include more information and references of these trends.**

1. Katzelnick LC, *et al.* Antigenic evolution of dengue viruses over 20 years. *Science*. 2021;374(6570):999-1004.
2. Zhang C, *et al.* Clade replacements in dengue virus serotypes 1 and 3 are associated with changing serotype prevalence. *J Virol*. 2005;79(24):15123-15130.
3. Adams B, *et al.* Cross-protective immunity can account for the alternating epidemic pattern of dengue virus serotypes circulating in Bangkok. *Proc Natl Acad Sci U S A*. 2006;103(38):14234-14239.

*“As is expected for DENV, we found that different serotypes transitioned in and out of dominance in each location during the twelve years (Figure 4). **This is consistent with genotype replacement events, which occur when a previously dominant lineage is replaced by another related, but distinct, lineage (Katzelnick et al. 2021; Zhang et al. 2005; Adams et al. 2006).** Various theories have been proposed to explain these events, including natural selection, immune pressure, and population bottlenecks. During the earlier years, however, we did detect some patterns.”*

Page 13, lines 394 and 400. It would be useful to mention how large the Cuban and Jamaica DENV3-III clades are (e.g. number of sequences from Cuba and Jamaica in relation to total number of sequences belonging to each clade).

We **added this to the text** as follows:

*“Our phylogeographic analysis further shows that the large cluster of DENV-3 sequences from Cuba in 2022 (**148 Cuban sequences out of 150 total sequences in the clade, posterior support for location = 1.0**)...”*

*“Sequencing travel-associated infections also revealed an earlier but distinct introduction of this genotype into Jamaica (**all 11 sequences in the clade, posterior support for location = 1.0**)...”*

Page 13, lines 413 and 414. Please include 95% Bayesian credible intervals for the number of virus transitions obtained from phylogeographic analyses.

Due to the potential impacts of sampling biases brought up by the reviewers and that this part is not central to our story, we **removed the section (including the previous Figure 6B)** on general within-Caribbean spread. A more detailed explanation was provided in response to Reviewer 1.

Page 14, lines 421-423. The sink-source results obtained using phylogeographic analyses are sensitive to sampling biases (heterogeneity in genomic sequencing data availability across locations). I would suggest including a supplementary figure showing how representative are sequence counts in relation to the estimated number of infections in key considered locations.

This **section was removed** - see explanation above.

Page 19, lines 561–563: “We attempted to correct this by using numbers in our calculations by only using cases that would have been detected in those years using the standard practice”. Fix typo.

We **edited** the referenced sentence:

*“We attempted to correct this by ~~using numbers in our calculations by only using cases~~ **in our calculations** that would have been detected in those years using the standard practice (i.e. not using “travel to Cuba” as the reason for DENV testing since that is not common practice for travel from other locations).”*

Page 22, lines 677 and 678. Please check the formulas of the final model.

We apologize for this oversight. When we downloaded our Google Doc into Word, the formula went missing and we didn’t catch it. It has been **added back into this submission**.

Page 23, line 730. “70%” of genome coverage?

We **edited** the referenced sentence as suggested:

*“We used all sequences sampled from the Caribbean and then took up to 10 sequences per country or territory per year per serotype, with a strict coverage cut-off applied to all sequences of **70% of genome coverage**.”*

Reviewer 3

The paper is relatively well written, has a clear and substantive narrative and is of public health importance.

We thank the reviewer for their positive response and for the time to review our manuscript.

Mainly, I found the section on the phylogeographic analyses hard to parse – particularly what constitutes an introduction, why it was defined in that manner to exclude subtree structure below the internal node of interest and why the authors did not instead opt for a Markov Jump analysis to capture the uncertainty and timing of transitions (it is unclear if they did – I spotted a logger in their XML).

We define introductions based on the oldest transition from non-country of interest to country of interest in our dataset. We do this because while lineages may leave and re-enter the country, we are primarily interested in the original introductions. This is also related to why we do not use the Markov jump analysis, which does not provide a per-branch analysis and so is challenging to parse the first introduction into the country of interest. **We better defined what constitutes as an introduction in our revised methods:**

“We defined an introduction as the oldest node on a branch which transitioned into the territory/region of interest, and counted all downstream nodes as part of the same introduction regardless of further exports/reintroductions. ~~Introductions were allowed to leave the country or territory/region of interest: that is, all downstream nodes in the country or territory of the first transition node were assigned to the same introduction regardless of nodes~~”

~~outside of the region in between. The time of introduction was taken conservatively, as the time of the transition node itself, and should be interpreted as the latest approximate time that the introduction could have occurred (i.e. the introduction could have occurred earlier, but not later).~~"

Related to this, **we decided to remove the within-Caribbean phylogeographic section that was presented in the previous Figure 6B** so that we are not overstating within-region dynamics. This section has the most concerns about the timing of introductions and impacts of sampling biases, but it is not central to our story. A more detailed explanation was provided in response to Reviewer 1.

222: Puerto Rico is the only country classified in the upper category, with the other four in the lower category? Not sure this is powered for interpretation, and at least needs some caveating in text for the reader to contextualize the results.

This observation was based on 3 observations (for Puerto Rico, Dominican Republic and Jamaica). Due to the potential impact of the small sample size and that this part is not central to our story, we **removed this statement** on the impact of GDP on dengue case reporting.

~~"We also estimate that having a higher GDP is associated with a 111% more reported dengue cases (95% CI: 12%-314%), which supports that increased resources are associated with increased case-finding efforts."~~

226-237: Do these countries have similar abundances of viable vectors? Do they have similar mosquito control programs? How have the control programs changed in recent times and how do you expect this to influence incidence?

There is no public information available on mosquito abundances or control measures in these countries. With regards to vector abundance, the only way to estimate this data would be by modeling weather variables. We would expect infections in travelers to reflect local dynamics. For example, improved mosquito control programs/decreased viable vectors in a country should lead to fewer infected travelers. Therefore, we feel that these local dynamics should be accounted for in our model.

278: I'm assuming there is not high resolution spatial information on what serotype circulates where, and how that is reflected in travelers?

That is correct there is not within country spatial information on serotypes. For the most part, there is not even yearly data on the proportions of serotypes for comparison to our traveler data. The only location with these data are from Puerto Rico 2010-2020 (Ryff et al, Ref #37), and we confirmed in the Results that our trends generally match the reported local trends. Mostly, serotype data is reported yearly by PAHO as a binomial variable (present vs absent). Therefore, while countries might report to PAHO that they detected all 4 serotypes in a given year, this does not provide information regarding the proportion of each serotype, nor which serotype is most responsible for an outbreak.

282-83: What is this attributable to?

Also in response to a comment from Reviewer 2, **we added further information and references to describe the patterns of serotype replacements:**

*“As is expected for DENV, we found that different serotypes transitioned in and out of dominance in each location during the twelve years (Figure 4). **This is consistent with genotype replacement events, which occur when a previously dominant lineage is replaced by another related, but distinct, lineage (Katzelnick et al. 2021; Zhang et al. 2005; Adams et al. 2006). Various theories have been proposed to explain these events, including natural selection, immune pressure, and population bottlenecks. During the earlier years, however, we did detect some patterns.**”*

Line 350 onwards: Please see comments on Markov Jump analyses and clarity required for definition of an introduction below.

Please see our response to the first comment and to further comments below.

356-367: I'm slightly confused as to why this is framed as validation; Surely line 365-367 is a given?

While we might assume that local and travel sequences from a country would cluster together, this is not a guarantee, as travelers may have different behaviors than locals, leading to infections by different lineages. Thus, comparing methods is helpful to support our approach.

391: What is the posterior support for that state? And for Jamaica on line 400.

We have **added the high posterior support to the text for these states as follows:**

*“Our phylogeographic analysis further shows that the large cluster of DENV-3 sequences from Cuba in 2022 **(148 Cuban sequences out of 150 total sequences in the clade, posterior support for location = 1.0)**...”*

*“Sequencing travel-associated infections also revealed an earlier but distinct introduction of this genotype into Jamaica **(all 11 sequences in the clade, posterior support for location = 1.0)**...”*

391: This is about a year's worth of uncertainty – worth annotating the distribution around the node in the figure?

We have added this HPD to figure 6A as suggested and updated the legend as follows:

*“The times of the most common ancestor of the clades are taken conservatively as introduction times, **with 95% HPDs indicated in brackets, and are indicated by circles.**”*

411: Again, I'm not sure if these represent median Markov jumps across time (logger in XML?), or introductions as defined in Methods section, which I find unclear and am missing the custom python script from the github. Also, these have no credible (e.g. Markov jump) or confidence intervals – phylogenetic uncertainty should be accounted for - especially as you only include 500 trees - for or are all trees highly supported?

Due to the potential impacts of sampling biases brought up by the reviewers and that this part is not central to our story, we **removed the section that was presented in the previous Figure 6B** on general within-Caribbean spread.

411-: What are the timings of these introductions? Any interesting patterns in seasonality in source-sink?

We **removed this section** - please see above.

418: As these numbers are fundamentally subject to sampling biases as mentioned in 420, it would be good to have an idea of how many sequences from each location are in each serotype tree. Also worth discussing how you think your downsampling framework of 10/country/year per serotype would affect this (as it's as arbitrary as all downsampling schemes!), especially as you include all Cuban sequences.

We **removed this section** - please see above.

429: Need HPDs on introduction estimates – worth including in figure 6 A, C too.
430: Need posterior support values for the source locations (“likely from Puerto Rico”)

We have added posterior support and HPDs to the figures and text as suggested. We also note that we swapped the source locations for the two introductions in the original text. We have therefore edited the text as follows:

We estimate that they were introduced into Cuba by at least ~~in early 2016~~ (95% HPD 2015-12-20 to 2016-12-15, likely from Puerto Rico, posterior support = 0.77 ~~Central America or Mexico~~) and late 2017 (95% HPD 2017-02-20 to 2018-06-25, likely from Central America or Mexico, posterior support = 1.0 ~~Puerto Rico~~), respectively, and both continued to circulate through the 2019 and 2022 outbreaks.

431: Do you think their relative frequencies reflect sampling biases or causality? What is the spatial resolution of this co-circulation?

Since these are travelers and the only location data that we have is “Cuba”, we cannot speculate on the spatial resolution. We also cannot determine if the changes in frequency are due to differences in fitness, stochastic epidemiological effects, or sampling biases. To include this uncertainty we updated the referenced statement:

*“Their relative frequencies changed between 2019 and 2022, with clade B dominating in 2019 but clade A increasing to become more even in frequency in 2022, **though these trends may be impacted by sampling biases.**”*

491: Too strong and causal language – not proved or quantified.

We **edited** the referenced sentence:

*“Given the speed of these events, it is **possible** ~~likely~~ that this DENV-3 lineage has a **significant** transmission advantage over at least some DENV lineages in the Americas.”*

499: Is there a reference/context for immune waning of DENV for the reader?

We have **added additional context and references** (see below).

1. Aogo RA, *et al.* Effects of boosting and waning in highly exposed populations on dengue epidemic dynamics. *Sci Transl Med.* 2023;15(722):eadi1734.
2. López L, *et al.* Considering waning immunity to better explain dengue dynamics. *Epidemics.* 2022;41:100630.
3. Forshey BM, *et al.* Dengue Viruses and Lifelong Immunity: Reevaluating the Conventional Wisdom. *J Infect Dis.* 2016;214(7):979-981.
4. Salje H, *et al.* Reconstruction of antibody dynamics and infection histories to evaluate dengue risk. *Nature.* 2018;557(7707):719-723.
5. OhAinle M, *et al.* Dynamics of dengue disease severity determined by the interplay between viral genetics and serotype-specific immunity. *Sci Transl Med.* 2011;3(114):114ra128.
6. Waggoner JJ, *et al.* Homotypic Dengue Virus Reinfections in Nicaraguan Children. *J Infect Dis.* 2016;214(7):986-993.

*“It subsequently spread throughout Central America and the Caribbean, leading to widespread epidemics of dengue hemorrhagic fever^{41,43-45}. **DENV immune waning has previously been observed (Aogo et al. 2023; López et al. 2022; Forshey et al. 2016; Salje et al. 2018; OhAinle et al. 2011; Waggoner et al. 2016)** and the sudden DENV-3 re-emergence stemming from that introduction could be attributed to the loss of immunity during the 16-year hiatus, which could be similarly true for the recent DENV-3 introduction into Cuba.”*

512: What do you mean by susceptibility here? Do we think these lineages are limited in subpopulations with different levels of immunity? Do you have spatial data for these sequences within Jamaica?

By susceptibility, we refer to whether the patient has been exposed to DENV-3 previously and has developed homotypic immunity, or whether they lack immunity to this serotype. There is a possibility that these lineages could be limited to subpopulations within Jamaica; however, since this lineage has only been detected in travelers and we have no local sequences, this is impossible to determine at this time. We do not have information about where these patients traveled within Jamaica, so cannot provide spatial data for these sequences.

521: Might need expanding on what you mean by “population and environmental effects”

We **clarified** the referenced sentence:

*“This indicates synchronized outbreaks are more likely driven by population (e.g. **time since last outbreak**) and environmental effects (e.g. **El Niño years**) than virus-related factors.”*

609: How many patients had traveled to more than one location? How do you expect this exclusion to affect your estimates?

We **included this information in the methods section.**

*“For this study, we only included patients who traveled to one endemic location within the 2 weeks prior to symptoms onset so we could more accurately sort the temporal and spatial distribution of travel-associated cases. **This led to the exclusion of 28 patients who traveled to multiple countries during our study period, constituting 1.2% of the 2,300 total cases (Table S1).** Within the Caribbean, the focus of this paper, there were 18 patients that were excluded due to traveling to multiple countries, constituting 1% of the 1,815 total cases. Therefore, due to the small number of these cases, we did not feel that excluding them would affect our analysis. We aggregated the data by year and by location of likely exposure (i.e., travel origin).”*

677-678: The notation is not parseable in the equation unfortunately for my review. Additionally, there are several parameters that require explanation in text.

We apologize for this oversight. When we downloaded our Google Doc into Word, the formula went missing and we didn't catch it. It has been **added back into this submission.** All **parameters are now described in the methods.**

694: Why only those countries?

We only tested for autocorrelation with those three countries because the test was done as part of our model development, and the model was based on those countries.

742: Confusing phrasing – are these clock rates? Also worth mentioning the priors on the clock.

We **rephrased the referenced sentence** for clarification and with the priors:

*“The mean **clock rate** of each tree, **using a CTMC scale prior on the mean and an exponential prior on the standard deviation**, ~~weighted by the branch lengths and followed by~~ and their 95% HPDs are: 6.61×10^{-4} (5.81×10^{-4} to 6.41×10^{-4}), 8.20×10^{-4} (7.78×10^{-4} to 8.65×10^{-4}), 6.69×10^{-4} (6.28×10^{-4} to 7.05×10^{-4}) and 7.68×10^{-4} (6.34×10^{-4} to 8.94×10^{-4}) substitutions/site/year for DENV-1 to -4, respectively.”*

754: Were grid points subject to sensitivity testing?

We did not conduct sensitivity testing on the gridpoints as the selection of external gridpoints is standard when not asking questions or concerned about past population dynamics (see Hill and Baele, 2019).

758: 60% burn-in is extremely high for a chain – what were the ESS for these chains, and if they were extended e.g. doubled, were they still at equilibrium? Did the results for that dataset replicate over independent chains? 60% would suggest that a few underlying

parameters were fluctuating for the majority of the chain and may indicate issues with the model.

We agree that 60% is high for burn-in, but we find this happens a lot with clock models which are not simply a strict clock, such as the relaxed clock model we use here, especially those with high numbers of sequences (e.g. Hill et al., 2022). We ran at least two chains (following best practice) and they always converged to the same point for each of the serotypes and sufficient ESS values for the parameters of interest were reached.

760: Were the 500 trees sampled from the converged posterior at random? I'm assuming this was for computational tractability, but it is rather low – need to account for phylogenetic uncertainty in analyses.

Correct, the 500 trees were randomly sampled from the converged posterior. We **edited the text to include this information**. Sampling 500 trees is a common practice for these types of analyses (e.g. de Plessis et al., 2021. Science).

773: Why did the authors not log Markov jumps across the tree, which can capture location transitions and transition times far more accurately/easily – and is parsable by the TreeMarkovAnalyzer in the Beast tools suite. In one of the XMLs examined, there seems to be a Markov jump logger – why were introductions defined as per 773-777? For example “The time of introduction was taken conservatively, as the time of the transition node itself.” – no need for this as Markov jumps include their placement in time along the branch?

We **further refined our definitions for introductions in the Methods**, as described in previous comments above and to the next comment below, in part to account for sampling bias. In this case, the accuracy provided by Markov jumps may be misleading due to unsampled diversity along the branches. We therefore felt it was most appropriate to be conservative with the introduction times. We are also only interested in the first introduction along the branch (discussed in the answer below), and this is not measured using TreeMarkovHistoryAnalyzer (this produces a CSV with only the transition states and the height). The XML contains a Markov jump logger because when we began the analysis we were not sure whether we would use the Markov jumps in the post-hoc phase of the study, and so we included them pre-hoc in the interests of efficiency. We apologize for leaving the Markov jump logger in the XML as it may be confusing to the reader, and **we have thus removed it**.

773-777: I find this text very confusing – I'm not entirely sure what constitutes an introduction based on this. So, an introduction is only the first node in a lineage after a transition, with all subtending nodes assigned the same state? What if that state is poorly supported? What if there's true mixing in the lineage below it, which will be lost by assigning it the state of a node several nodes up? Again, phylogenetic uncertainty can be accounted for in Markov jump analyses across a large enough set of trees from the posterior.

We choose the earliest transition node on the branch to define the introduction, and then count downstream nodes as part of the same introduction, and we **rephrased the text as follows to hopefully make this clearer**:

“Introductions were inferred using a custom Python script. We defined an introduction as the oldest node on a branch which transitioned into the territory/region of interest, and counted all downstream nodes as part of the same introduction regardless of further exports/reintroductions.”

We do this to be conservative with the conclusions that we are drawing from the dataset which, as noted, contains sampling bias. We are also less interested in the downstream exports and reintroductions as these are likely to be driven by sample numbers in different countries, rather than the more conservative option which we have chosen here.

In light of this, we also **removed the within-Caribbean dynamics section that was shown in the previous Figure 6B**, as this is most affected by our decision to define introductions such as this. We have also made it clear that our introduction timings are the latest time that a branch is in the country/territory.

Of all of the introductions we define this way, only one transition node has a posterior support of less than 0.5. This is a DENV2 introduction which we do not discuss in the paper, with a posterior support of 0.42 of having Jamaica as a state. We note, however, that the next state with the highest support is Puerto Rico (0.34) so we are still confident in the introduction coming into the Caribbean at that time.

773: Also I may be looking past it, but I do not see the custom python script in the github, so I can't figure out how introductions were parsed.

We apologize for this oversight and have now **added the script into the github**.

167 + 173: What does the inclusion of enhanced syndromic surveillance case numbers do to your estimates for Cuba?

We **included this information in the methods section**.

“When we compared the local and travel case trends for Cuba with versus without the enhanced surveillance, we found that excluding these travel cases led to decreased correlation ($R = 0.575$ with $p = 0.032$ versus $R = 0.496$ with $p = 0.070$, respectively). When we compared the travel and local infection rates for Cuba with versus without the enhanced surveillance, we once again found that case exclusion led to decreased correlation ($R = 0.516$ with $p = 0.059$ versus $R = 0.458$ with $p = 0.1$, respectively). Therefore, we determined that including all the data would increase our estimates of the outbreak sizes in Cuba, and chose to use only cases captured by traditional reporting to not overestimate these local outbreaks.”

179: Might be worth discussing, in context of the rest of the world, why Florida in particular is a good sentinel for Cuba. It appears in the Discussion around 467, but should be set-up in the Introduction.

We **included this information in the introduction**.

*“In Florida, the number of travel-associated dengue cases has increased dramatically in recent years ³¹. **Florida is a good sentinel for the Caribbean, due to its geographic location and high volume of travel back and forth between Florida and the islands.** Therefore, we hypothesize that we can use surveillance of dengue-infected travelers diagnosed in Florida who recently returned from the Caribbean to better reconstruct DENV dynamics in the region.”*

198: Figure 2 “travel volume to Florida from 2020-2020” – assuming the end date is a typo?

We **corrected the typo** in the legend.

*“The negative correlation between the local and travel infection rates may have been driven by a decreased travel volume to Florida from 2020-~~2020~~**2022**”*

392: Are these sequences in Figure 6A?

We apologize for the confusion with this statement. The sequences and results are in the referenced articles, in which we included an additional reference that highlights the introduction of this lineage into the US. We **updated the statement for clarity as follows.**

*“~~Using data from the FDOH and the CDC Dengue Branch, w~~**We and others** found that this lineage has already spread to Puerto Rico, Florida, ~~and~~ Arizona, **and Brazil** by 2022, ~~and others recently detected it in Brazil~~^{33,39}.”*

477: Perhaps a bit overstated – not “factors”, just serotype.

We **edited the referenced sentence** as suggested:

*“To investigate the **DENV serotypes** ~~factors~~ responsible for these outbreaks, we analyzed serotype data from infected travelers returning to Florida...”*

519: “not one lineage”?

We **edited the referenced sentence** as suggested:

*“Even in outbreak years, such as 2019 or 2022, there was ~~no~~ **not** one lineage that became dominant in the region...”*

623: Might be a phrasing problem but 397/929 cases in 2022 does not warrant the applied “Only”

We **edited the referenced sentence** as suggested:

*“~~Only 18~~ **Eighteen** of the 413 travel-associated cases in 2019 and 397 of the 929 cases in 2022 were first identified via syndromic surveillance and used in our analysis.”*

625: “Without this additional intervention” – assuming it refers to the addition of travel history to Cuba, but might want to rephrase for clarity.

We **rephrased the referenced sentence** for clarity as follows:

*“Of note, 116 cases identified via enhanced surveillance in 2022 would not have met **our** case criteria **definition** without this additional **criteria** intervention.”*

REVIEWERS' COMMENTS

Reviewer #1 (Remarks to the Author):

The authors addressed all my comments

Reviewer #1 (Remarks on code availability):

I have looked at the code, w/o trying to run it. The BEAST input files seem to be available, so are the R scripts and python notebooks to recreated figures. The BEAST output files to run the notebook are missing and the README doesn't contain any information

Reviewer #3 (Remarks to the Author):

All points raised have been adequately addressed.